# Maternal and perinatal obesity induce bronchial obstruction and pulmonary hypertension via IL-6-FoxO1-axis in later life

Jaco Selle [1,16], Katharina Dinger[1,2,16], Vanessa Jentgen[1,16], Daniela Zanetti [3,4], Johannes Will[1], Theodoros Georgomanolis [5], Christina Vohlen [1,6,7], Rebecca Wilke[1], Baktybek Kojonazarov[7], Oleksiy Klymenko[7], Jasmine Mohr[1], Silke v. Koningsbruggen-Rietschel [8], Christopher J. Rhodes [9], Anna Ulrich [9], Dharmesh Hirani [1,2,7], Tim Nestler [10], Margarete Odenthal [2,10], Esther Mahabir [11], Sreenath Nayakanti [12], Swati Dabral [12], Thomas Wunderlich[2,13,14], James Priest [3], Werner Seeger [7,12,15], Jörg Dötsch [6], Soni S. Pullamsetti [7,12,15] & Miguel A. Alejandre Alcazar [1,2,7,14,15 ✉]

Obesity is a pre-disposing condition for chronic obstructive pulmonary disease, asthma, and pulmonary arterial hypertension. Accumulating evidence suggests that metabolic influences during development can determine chronic lung diseases (CLD). We demonstrate that maternal obesity causes early metabolic disorder in the offspring. Here, interleukin-6 induced bronchial and microvascular smooth muscle cell (SMC) hyperproliferation and increased airway and pulmonary vascular resistance. The key anti-proliferative transcription factor FoxO1 was inactivated via nuclear exclusion. These findings were confirmed using primary SMC treated with interleukin-6 and pharmacological FoxO1 inhibition as well as genetic FoxO1 ablation and constitutive activation. In vivo, we reproduced the structural and functional alterations in offspring of obese dams via the SMC-specific ablation of FoxO1. The reconstitution of FoxO1 using IL-6-deficient mice and pharmacological treatment did not protect against metabolic disorder but prevented SMC hyperproliferation. In human observational studies, childhood obesity was associated with reduced forced expiratory volume in 1 s/forced vital capacity ratio Z-score (used as proxy for lung function) and asthma. We conclude that the interleukin-6-FoxO1 pathway in SMC is a molecular mechanism by which perinatal obesity programs the bronchial and vascular structure and function, thereby driving CLD development. Thus, FoxO1 reconstitution provides a potential therapeutic option for preventing this metabolic programming of CLD.

A full list of author affiliations appears at the end of the paper.

Obesity, a condition characterized by an increase in white adipose tissue (WAT), metabolic disorders, and an increased risk of cardiovascular and pulmonary diseases, is associated with increasing socio-economic burden[1–5]. Studies have identified chronic subacute inflammation consequent to the secretion of adipocytokines from WAT as a central mechanistic factor underlying the pathogenesis of obesity-related diseases[6–8]. Obesity itself remains a complex trait attributed to both genetic and environmental factors despite marked progress in the treatment of the associated metabolic syndrome and cardiovascular and pulmonary diseases.

Recent studies have identified maternal obesity as a determinant of the risks of obesity, metabolic disorders, and cardiopulmonary diseases in offspring[3,9–13]. Particularly, the incomplete pulmonary development at birth renders the lung vulnerable. As a result, the perinatal window is a highly susceptible period regarding the risks for chronic lung disease (CLD) beyond infancy, and structural and functional changes similar to those observed in adults with chronic obstructive pulmonary disease (COPD) or cardiovascular diseases may occur[14–20]. This concept of perinatal metabolic imprinting, enhancing the susceptibility to chronic health disorders, has been described as metabolic programming over the last decades[21]. However, the underlying molecular mechanisms remain elusive. The identification of signaling pathways that link obesity with chronic cardiovascular and pulmonary diseases, which could serve as therapeutic targets, is urgently needed.

Obesity and chronic cardiopulmonary diseases converge upon the same inflammatory signaling pathways[6,22,23]. WAT, the largest endocrine organ, secretes a variety of inflammatory adipocytokines[6,7]. Of these, IL-6 is particularly implicated in inflammation-driven processes that lead to obesity-related diseases[22,24]. Moreover, an increase in circulating levels of IL-6 is closely linked with chronic cardiopulmonary diseases such as COPD and pulmonary arterial hypertension (PAH)[25–31]. Both diseases are characterized by proliferative processes and phenotypic features such as bronchial and vascular wall remodeling with smooth muscle cell (SMC) proliferation[32–34], and anti-inflammatory treatment has been considered in both the preclinical and clinical disease forms[23,35,36]. Thus, exposure to adipocytokines during a critical developmental window could adversely affect bronchial and lung vascular development and predispose the individual to a life-long CLD.

FoxO proteins comprise a subgroup of the Forkhead family of transcription factors and are characterized by a conserved DNA-binding domain[37]. One family member, FoxO1, not only plays an important role in metabolic processes, including insulin signaling and adipogenesis but also regulates cell homeostasis and cell cycle progression and has been implicated in vascular remodeling and PAH pathogenesis[34,38,39]. The activities, subcellular localization, and stability of FoxO proteins are controlled by post-translational modifications such as phosphorylation[40,41]. In the nucleus, FoxO1 can act as both a transcriptional activator and repressor. Pharmacological approaches to IL-6 inhibition and FoxO1 modulation provide novel and clinically-relevant strategies for the prevention of a developmental predisposition to CLDs. To date, however, the potential obesity-FoxO1 axis has not been explored in the context of the developmental origin of CLD.

In this study, we aimed to determine whether the blockade of IL-6 signaling and targeting of FoxO1, as a central downstream effector on which proliferative and inflammatory signaling are assumed to converge, might offer an avenue for interfering with the metabolic programming of CLD and thus provide a potential preventive strategy. We demonstrate that perinatal exposure to a maternal high-fat diet (HFD) and subsequent obesity during early life induce bronchial and vascular SMC hyperproliferation, which

are key characteristics of bronchial and vascular remodeling in COPD and PAH, respectively. We provide evidence for an adipocytokine-lung axis mediated by FoxO1, underlying bronchial and vascular developmental abnormalities with the subsequent emergence of bronchial obstruction and PAH, offering potential preventive concepts.

## Results

**Perinatal HFD induces transient early-onset obesity and elevated adipocytokine levels in offspring.** We established a murine model of metabolic programming to study the impact of perinatal HFD and maternal obesity on offspring, as depicted in Fig. 1A. Notably, consumption of a HFD for 7 weeks before mating induced overweight (Fig. 1B) with progressive weight gain (Fig. 1C) as well as impaired glucose tolerance (Fig. 1D), indicative of insulin resistance in virgin female mice. After mating, dams in the maternal HFD (HFD$^{mat}$) and standard diet (SD$^{mat}$) groups were maintained on their respective diets during gestation and lactation. The offspring of these dams are hereafter referred to as HFD and SD, respectively.

Both groups of offspring were studied at two time points: the early phase at postnatal day 21 (P21; acute metabolic impact) and the late phase at P70 (long-term sequelae). At P21, the HFD group exhibited overweight and a two- to threefold increase in epigonadal WAT relative to the body weight, along with increased WAT-specific expression of the adipocytokine genes *Il6* (up to 5–8-fold) and *Lep* (leptin; up to 15-fold) relative to the SD group (Fig. 1E). A chronic subacute inflammatory state caused characteristics of insulin resistance, a hallmark of obesity. Measuring the serum concentrations of IL-6 and leptin and assessing glucose tolerance in the offspring, elevated serum concentrations of these adipocytokines were indeed detected, associated with impaired glucose uptake in HFD relative to SD (Fig. 1F). After weaning at P21, both groups received SD chow and were followed up until P70. In the HFD group, the mice exhibited a decrease in body weight. However, the WAT relative to body weight remained higher in the HFD group than in the SD group. Both groups exhibited similar *Il6* and *Lep* mRNA expression levels and no significant difference in the serum concentrations of the corresponding adipocytokines. Accordingly, the glucose tolerance was not significantly different in the HFD and SD at P70 (Fig. 1G, H). Along with these findings, prior studies from our group showed hyperinsulinemia in HFD at P21, and no differences in serum insulin between HFD and SD at P70.[42] Gene expression and the serum levels of other adipocytokines such as adiponectin and IL-1β were reduced or not altered, respectively, in HFD at P21 and P70 when compared to SD (Supplementary Fig. 1A–D). In summary, perinatal HFD with maternal obesity induces early postnatal obesity with transiently elevated adipocytokine levels of IL-6 and leptin along with impaired glucose metabolism in the offspring.

**Perinatal HFD is linked to bronchial and vascular remodeling, bronchial obstruction, and pulmonary arterial hypertension.** The above-described coupling of a transient adipocytokine-linked inflammatory state with impaired metabolism enabled an investigation of the long-term impacts of these effects on the bronchi at P70. First, we stained lung sections for α-smooth muscle actin (SMA), a SMC marker, and evaluated the thickness of the bronchial smooth muscle layer (bSML). Notably, we observed a significant increase in the bSML in bronchi with diameters <150 μm in HFD relative to SD, and this increase was related to an increase in airway resistance in HFD when compared to SD (Fig. 2A–C). Interestingly, the response to increasing dosages of methacholine, a bronchoconstrictor, confirmed increased airway

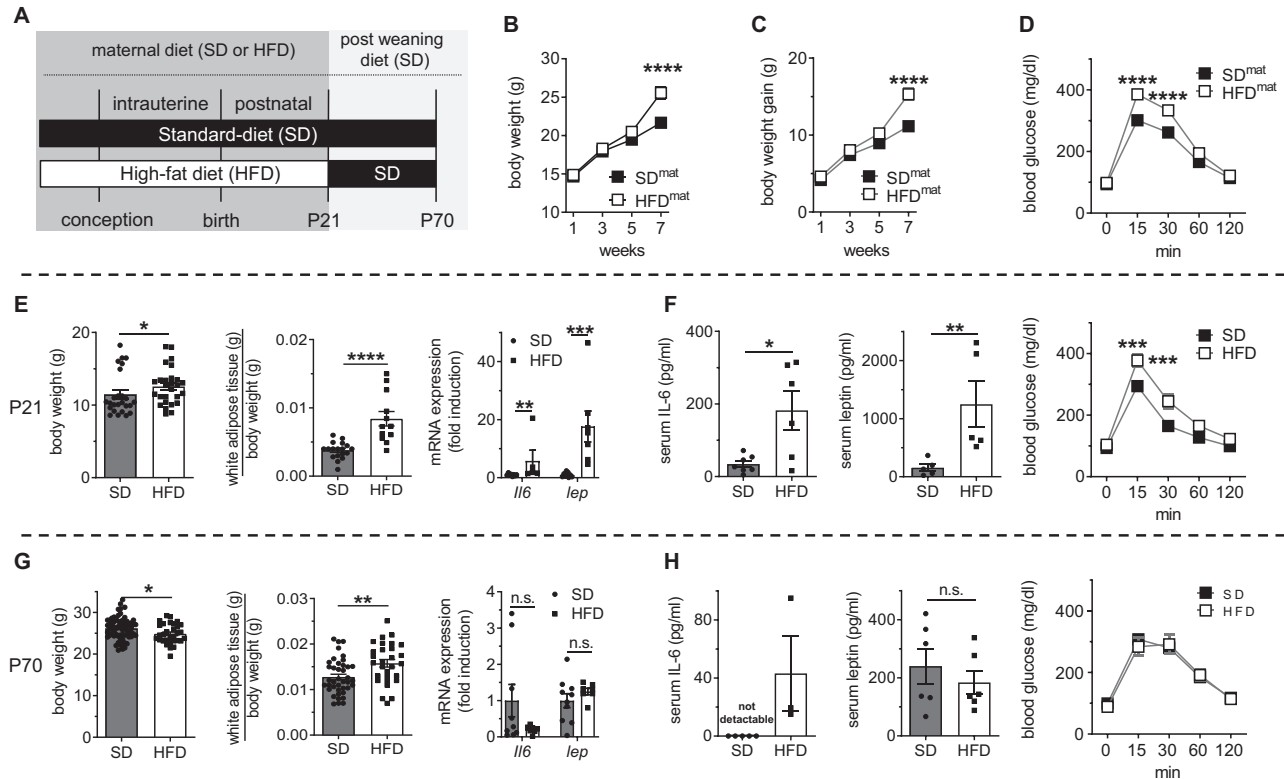

**Fig. 1 A maternal prenatal and perinatal high-fat diet (HFD) and metabolic disorder induce transient early-onset obesity, elevated adipocytokine levels, and impaired glucose tolerance in offspring. A** Animal model of metabolic programming. Female mice were fed a HFD for 7 weeks prior to mating; control mice received a standard diet (SD). The dams remained on their respective diets throughout gestation and lactation. The offspring of both groups were weaned at postnatal day 21 (P21) and fed SD until P70. **B–D** Body weight (**B**), body weight gain (**C**), and intraperitoneal glucose tolerance test (ipGTT) results (**D**) of dams prior to mating. **E** The body weight, ratio of epigonadal white adipose tissue (WAT) relative to body weight and mRNA expression of *Il6* (interleukin-6) and *Lep* (leptin) in the WAT of offspring at P21 using qRT-PCR; HFD relative to SD. **F** Serum IL-6 and leptin concentrations in the offspring were assessed by ELISA at P21; offspring were subjected to the ipGTT. **G** The body weight, ratio of WAT relative to the body weight and expression of *Il6* and *Lep* in the WAT of offspring at P70 using qRT-PCR; HFD relative to SD. **H** Serum IL-6 and leptin concentrations in the offspring at P70 using ELISA. Offspring were subjected to the ipGTT at P70. Data are shown as mean ± standard error of the mean **B**: SD $n = 32$, HFD $n = 40$; **C** SD $n = 32$, HFD $n = 40$; **D** SD $n = 36$, HFD $n = 49$; **E** SD $n = 9–25$, HFD $n = 7–27$; **F** SD $n = 5–30$, HFD $n = 5–7$; **G** SD $n = 10–68$, HFD $n = 7–31$; **H** SD $n = 6–21$, HFD $n = 3–7$. Statistical analyses were performed using the two-sided Mann–Whitney test, two-sided Student's *t* test or two-way ANOVA followed by the Bonferroni post-test; *$P < 0.05$; **$P < 0.01$; ***$P < 0.001$; ****$P < 0.0001$. Gray = standard diet; white = high-fat diet. A detailed list of sample sizes and *P* value per graph is provided in the Supplemental Material. Source data are provided as a Source Data file.

resistance at baseline but did not indicate a bronchial hyper-reagibility in HFD relative to SD (Supplementary Fig. 2A, B). Second, we investigated whether perinatal HFD might also adversely affect the small pre-capillary vessels. Here, we observed a significant increase in partially and completely muscularized microvessels in the lungs of HFD in comparison to SD, which was supported by a significant increase in the medial wall thickness of microvessels with diameters of 20–70 and 70–150 μm (Fig. 2D–F). The number of microvessels (0–20 μm and 20–100 μm) were not different between HFD and SD at P70 (Supplementary Fig. 2C). Echocardiographic analysis revealed an increased heart rate, marked reduction of cardiac output (CO) and stroke volume (SV), elevated right ventricular internal diameter (RVID), and decreased tricuspid annular plane systolic excursion (TAPSE) (Fig. 2G–K). The right ventricular systolic pressure (RVSP) was not significantly altered in HFD when compared to SD, possibly due to the inability of the RV to maintain pressure. However, the total pulmonary vascular resistance index (TPVRI) was significantly increased in HFD mice when compared to SD (Fig. 2L, M). Left ventricular (LV) ejection fraction (EF), assessed by echocardiography, did not significantly differ between HFD and SD (Fig. 2N). In addition, systolic blood pressure (SBP) and diastolic blood pressure (DBP) were measured

by invasive catheterization and were similar in both groups HFD and SD, indicating that systemic hemodynamic was not affected (Fig. 2O, P). Moreover, we found that the hemodynamic abnormalities of the right ventricular system after perinatal HFD were accompanied by increased expression of *Nppa*. This gene encodes an atrial natriuretic peptide, an indicator of high right ventricular pressure (Fig. 2Q).

To further support the notion that the bronchial and vascular changes after perinatal HFD are linked to an early increase in lung SMC expression at P21, we identified a greater abundance of αSMA protein in HFD relative to SD lungs using immunoblotting (Fig. 2R). Finally, we extracted RNA from bronchi and microvessels isolated via laser microdissection (LMD) at P21 and measured the expression of *Acta2* as an indicator of SMC using qRT-PCR. Our analysis revealed a twofold increase in the expression of the gene encoding *Acta2* in HFD relative to SD (Fig. 2S). In addition, immunohistochemical staining and quantification of specific T-cells, leucocytes (B cells), and neutrophils in the lung using CD3, CD45, and Ly6G as respective markers did not show differences in the number of immune cells between HFD and SD at P21 and P70, supporting the notion that the inflammatory response in the lung after perinatal obesity is likely related to WAT (Supplementary Fig. 3A–C). In summary,

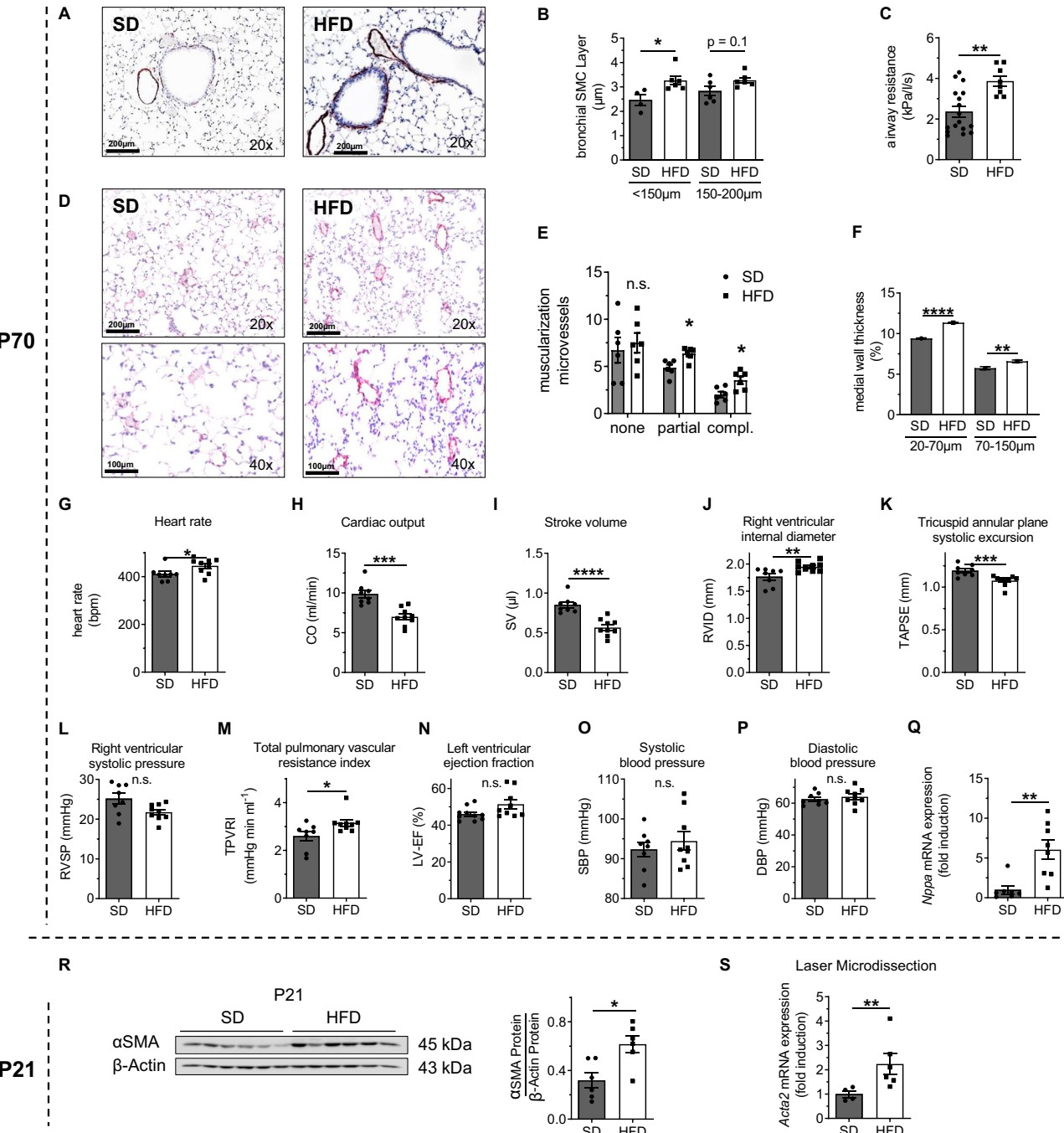

**Fig. 2 A maternal high-fat diet (HFD) is associated with thickening of the bronchial and vascular smooth muscle cell (SMC) layer as well as bronchial obstruction and pulmonary arterial hypertension beyond infancy. A** Representative images of the lungs of offspring at postnatal day 70 (P70). Immunohistochemistry for α-smooth muscle actin (αSMA) was performed to assess the bronchial SMC layer in the offspring of dams fed a HFD or standard diet (SD). **B** Quantitative measurement of the bronchial SMC layer in bronchi with diameters <150 μm and 150–200 μm. **C** Assessment of respiratory airway resistance using body plethysmography at P70. **D–F** Representative images of the lungs of offspring at P70. Dual staining for αSMA and von Willebrand factor (vWF) (**D**) was used to assess the muscularization of microvessels (20–100 μm) (**E**) and the medial wall thickness of microvessels (diameters: 20–70 μm and 70–150 μm) (**F**). **G–K** Echocardiographic assessment of heart rate (**G**), cardiac output (CO, **H**), stroke volume (SV, **I**), right ventricular internal diameter (RVID, **J**) and tricuspid annular plane systolic excursion (TAPSE, **K**). **L, M** Hemodynamic measurement of right ventricular systolic pressure (RVSP, **L**), and total pulmonary vascular resistance index (TPVRI, **M**). **N** Echocardiographic assessment of the left ventricular (LV) ejection fraction (EF). **O, P** Measurement of the systolic blood pressure (SBP) and diastolic blood pressure using invasive catheterization. **Q** qRT-PCR assessment of *Nppa* expression in the heart at P70. **R** Immunoblot analysis of αSMA protein in total lung homogenates at P21; β-actin was used as a loading control. A summary of the αSMA abundance relative to β-actin is displayed next to the immunoblot. **S** qRT-PCR assessment of *Acta2* expression in laser-microdissected bronchi and vessels at P21; 18 s rRNA served as a housekeeping gene. Data are shown as mean ± standard error of the mean. **B, C, E–S** SD $n = 4$–17, HFD $n = 6$–9. Statistical analyses were performed using the two-sided Mann–Whitney test or two-sided Student's $t$ test; *$P < 0.05$; **$P < 0.01$; ***$P < 0.001$; ****$P < 0.0001$. Gray = standard diet; white = high-fat diet. A detailed list of sample sizes and $P$ value per graph is provided in the Supplemental Material. Source data are provided as a Source Data file.

our data demonstrate that maternal obesity induces bronchial and vascular remodeling with SMC hyperproliferation in the offspring, leading to bronchial obstruction and increased pulmonary vascular resistance.

**Transcriptomic analysis identifies the regulation of FoxO1 in the bronchi and vessels of offspring exposed to a perinatal HFD.** To further delineate key molecular mechanisms mediating SMC hyperproliferation in the offspring of obese dams, we isolated bronchi and vessels from lungs at P21 via LMD, extracted RNA, and performed a RNA sequencing analysis. The observed differences in gene expression between HFD and SD are illustrated in a heatmap and volcano plot in Fig. 3A, B. Next, we used CiiiDER software, a tool for predicting transcription factor binding sites, to perform a gene promotor analysis of the top 100 upregulated and downregulated genes with adjusted $P$ values <0.01, using a deficit value of 0.1[43]. FoxO1 binding sites were identified in 171 of the genes (85.5%; Fig. 3C), and 114 and 105 genes of these genes also contained putative binding sites for STAT3 and JUN, respectively, which are key effectors of IL-6 signaling. Interestingly, a further analysis of putative binding sites for other transcription factors, such as PPARγ or NFE2L2, revealed only 38 and 12 genes, respectively (Fig. 3D). We further observed the downregulation of the expression of the genes encoding *Foxo1* in HFD relative to SD (Supplementary Fig. 4). The RNA sequencing data were further confirmed using qRT-PCR and immunoblot. *Stat3* mRNA as well as *Socs3* mRNA as a downstream target of STAT3 signaling[44] were increased in bronchial SMC[HFD] (bSMCs[HFD]) when compared to bMSC[SD] and total lung STAT3 protein abundance was greater in HFD than SD; in contrast, Jun was not significantly altered (Fig. 3E, F). These findings suggest a dysregulation of STAT3 and FoxO1 signaling that may well contribute to the bronchial and vascular phenotypic changes in response to maternal obesity.

We next conducted a gene ontology analysis to identify the relevant underlying biological processes (Fig. 3G). Interestingly, the ten most significantly regulated biological processes (BP) involved angiogenesis, respiratory tube development, and vasculogenesis (left panel). Furthermore, a KEGG pathway annotation analysis of differentially regulated genes associated with HFD identified a number of relevant pathways, most notably hypertrophic cardiomyopathy, pro-angiogenic apelin signaling, vascular smooth muscle contraction, and dilated cardiomyopathy (DCM; Fig. 3G, right panel). In summary, perinatal HFD leads to aberrant vascular and bronchial development, which is possibly mediated by FoxO1 signaling.

**Perinatal HFD reduces FoxO1 expression and activates proliferation in bronchial and vascular SMC.** The preceding results demonstrate that perinatal HFD not only induces a transient metabolic disorder and obesity in offspring but also promotes bronchial and vascular remodeling with increased SMC layer thickness. This latter finding suggests early postnatal hyperproliferation of lung SMC in HFD. Together with the transcriptomic analyses, these observations prompted us to determine whether perinatal HFD induces SMC proliferation by dysregulating STAT3 and FoxO1 signaling. IL-6 is a key activator of STAT3, and HFD exhibited elevated serum IL-6 concentrations at P21. Therefore, we measured STAT3 showing a up to twofold higher level of phosphorylated STAT3 in the lungs of HFD relative to SD (Supplementary Fig. 5A). The activation of STAT3 in lung SMCs was confirmed by immunofluorescence co-staining for pSTAT3 and αSMA as a marker for SMCs showing an up to twofold increase of pSTAT3 in vascular and bronchial SMCs of lungs in HFD when compared to SD (Fig. 4A, B). To determine if other

pathways downstream of adipocytokines or insulin are activated in the lungs of HFD offspring at P21 we assessed phosphorylation of JNK and P38. These pathways were not different between HFD and SD (Supplementary Fig. 5B, C). Since HFD offspring exhibited glucose intolerance and hyperinsulinemia[42], we assessed AKT signaling in the lungs. While phosphorylated AKT was not different in lung homogenates between HFD and SD (Supplementary Fig. 5D), co-staining for pAKT and αSMA showed a compartment-specific twofold increase of pAKT in vascular and bronchial SMCs of HFD lungs when compared to SD (Fig. 4C, D). Next, we performed dual immunostaining for αSMA and Ki67, a proliferation marker, to assess SMC proliferation in the bronchi and microvessels and observed significantly more proliferating bronchial and vascular SMC in HFD than in SD (Fig. 4E, F).

Previous studies linked the cytoplasmatic sequestration and degradation of FoxO1 with cell proliferation[34]. Therefore, we stained lung tissues of mice after maternal and perinatal obesity (HFD) as well as control mice (SD) at P21 to detect both FoxO1 and αSMA. Subsequently, we assessed the cytoplasmatic and nuclear intensity of FoxO1 in bronchial and vascular SMC and observed a 90% reduction in FoxO1 staining and an increased nuclear-to-cytoplasmatic shift of this transcription factor in bSMC in HFD relative to SD (Fig. 4G–I). Interestingly, the abundance of FoxO1 in vascular SMC was not lower in HFD relative to SD. The patterns of nuclear-to-cytoplasmatic FoxO1 sequestration were similar in vascular SMC and bSMC, indicating decreased nuclear availability of FoxO1 in both SMC types in HFD (Fig. 4J–L).

We next assessed the expression of gene encoding *Foxo1*, *Foxo3*, *Foxo4*, and *Foxo6* in total lung homogenate using qRT-PCR and did not determine any changes in expression between HFD and SD at P21 or P70 (Supplementary Fig. 6A, B). To address a compartment-specific effect, we measured gene expression of FoxOs and target genes in the bronchi and microvessels after LMD at P21. Consistent with the loss of nuclear FoxO1 in SMCs, we observed a reduction in *Foxo1* mRNA as well as in *Cdkn1b*, *Bcl6*, and *Faslg*, and a mild increase in *Ccnb1* mRNA. Gene expression of *Foxo3*, *Foxo4*, and *Foxo6* was not significantly altered in HFD when compared to SD at P21 (Fig. 4M, N). Finally, we isolated primary lung SMC and demonstrated that perinatal HFD led to a significant reduction in the expression of *Foxo1* as well as its target genes *Bcl6* and *Gadd45a*; *Foxo4* was also slightly reduced, whereas *Foxo3* and *Fox6* were not changed (Fig. 4O, P). In summary, perinatal HFD and the associated transient metabolic phenotype represented by IL-6 elevation induce proliferative STAT3 and AKT signaling coupled with FoxO1 inactivation in bronchial and lung microvascular SMC.

**IL-6-FoxO1 regulates primary murine and human bronchial SMC (bSMC).** Having documented that perinatal HFD with maternal obesity causes elevated serum IL-6 concentration and STAT3 activation in lungs, along with nuclear FoxO1 exclusion as well as SMC hyperproliferation, we next isolated primary bSMC from lungs of 21-day-old mice and exposed the cells to IL-6+sIL-6R (IL-6 + R) to investigate IL-6-STAT3-FoxO1 signaling. Exposure of bSMC to IL-6 + R for 30 min markedly activated STAT3 and reduced as well as increased the nuclear and cytoplasmatic pFoxO1, respectively (Fig. 5A, B). Phosphorylation of FoxO3a, however, was not changed by IL-6 + R, indicating a possible FoxO1-specific effect of IL-6 in bSMC (Supplementary Fig. 7). After treatment of bSMC with IL-6 + R for 24 h, total FoxO1 was reduced in the nucleus and increased in the cytoplasm (Fig. 5C). The adipocytokines IL-1β and leptin, however, did not

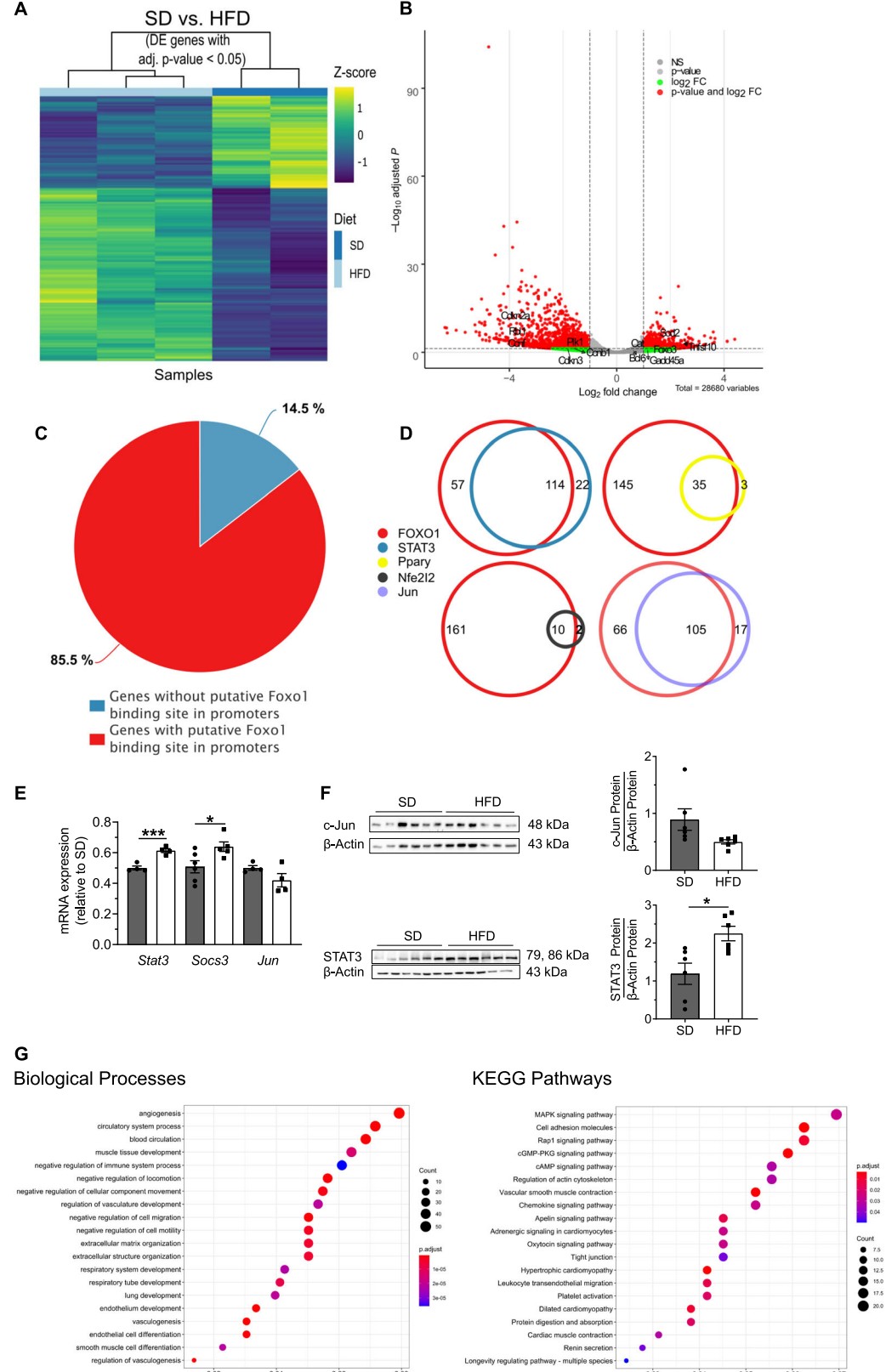

regulate cellular FoxO1 abundance or proliferation of bSMCs significantly (Supplementary Fig. 8A, B). To demonstrate that this IL-6-FoxO1 axis requires STAT3 we next exposed bSMC to Stattic, a STAT3 inhibitor, and determined a significant increase of total nuclear FoxO1 (Fig. 5D). Moreover, inhibition of STAT3 in bSMC using Stattic prevented the IL-6 + R-mediated reduction

of nuclear pFoxO1, the increase of cytoplasmatic pFoxO1, and the decrease of total nuclear FoxO1 abundance (Fig. 5E). We next demonstrated that IL-6 + R increases proliferation (cell count), whereas Stattic reduced viability of bSMC by 50 % (Fig. 5F). We next tested if activation of FoxO1 reverses the IL-6 effects on bSMCs. To this end, we assessed viability, proliferation, and

**Fig. 3 Identification of FoxO1-STAT3 signaling in bronchi, microvessels, and adjacent lung tissue from offspring at postnatal day 21 (P21). A** The heatmap from a transcriptomic analysis includes only genes with an absolute log-fold change (FC) > 2 (adjusted $P$ value <0.05); standard diet (SD) and high-fat diet (HFD) represent the offspring of standard diet (SD)-fed dams ($n = 2$) and HFD-fed dams ($n = 3$), respectively. Z-Score is indicated in a color score. **B** Volcano plot analysis; genes with an absolute FC > 2 and adjusted $P$ value <0.05 are indicated in red; genes with an FC > 2 but adjusted $P$ value >0.05 are green (DEGs in SD compared to HFD). Labels in the volcano plot indicate cell cycle and FoxO1 target genes. $P$ values are calculated using Wald test and $P$-adjusted values using the FDR/Benjamini–Hochberg approach. **C**, **D** Venn diagrams depicting the putative binding sites for the transcription factors Foxo1, STAT3, Jun, Nfe2l2, and Pparγ in the promoters of the top 100 upregulated and downregulated genes with adjusted $P$ values <0.01. **E** Gene expression of *Stat3*, *Socs3*, and *Jun* in bronchial smooth muscle cells (bSMCs) isolated from offspring of SD- and HFD-fed dams at P21 ($n = 4$/group); *Gapdh* served as the housekeeping gene. **F** immunoblot for c-Jun and total STAT3 in total lung homogenates from HFD and SD at P21 ($n = 6$/group); a densitometric summary of the STAT3 and c-Jun data relative to β-actin is displayed next to the respective immunoblot. **G** A functional enrichment analysis to identify the upregulated biological processes (SD vs HFD) associated with differentially expressed genes (DEGs) with adjusted $P$ values <0.05. A KEGG pathway analysis was used to identify the top ten upregulated pathways in the bronchi (SD vs HFD), microvessels, and adjacent lung tissue at P21.
**E**, **F** Data are shown as mean ± standard error of the mean. The statistical analysis was conducted using the two-sided Mann–Whitney test and two-sided Student's $t$ test; *$P$ < 0.05; ***$P$ < 0.001. Gray = standard diet; white = high-fat diet. A detailed list of sample sizes and $P$ value per graph is provided in the Supplemental Material. Source data are provided as a Source Data file.

apoptosis using MTT, BrdU, and Caspase3/7 assays, respectively. The analysis revealed that employing Paclitaxel-mediated activation of FoxO1 reversed the IL-6 + R-mediated increase in viability and proliferation, and the reduction of apoptosis in bSMCs (Fig. 5G). In addition, we show that the treatment of bSMC with Paclitaxel induces the expression of FoxO1 target genes (e.g., Bcl6 and Gadd45a; Supplementary Fig. 8C). Since we found hyperinsulinemia together with active AKT signaling in bronchial and vascular SMC in HFD, we next exposed bSMC to insulin. Insulin-induced phosphorylation of AKT was related to a nuclear-to-cytoplasmatic shift of pFoxO1, similar to the effect seen by IL-6-STAT3 signaling (Fig. 5H, I). Moreover, we found that IL-6 + R does not only activate STAT3, but also AKT signaling, and identified a STAT3-AKT interaction in bSMC (Fig. 5J). To further confirm that STAT3-mediated FoxO1 activation is AKT-dependent, we exposed bSMC from WT mice to IL-6 + R and AKT inhibitor and found that the inhibition of AKT prevents IL-6-mediated reduction of nuclear and increase of cytoplasmatic pFoxO1 (Fig. 5K). To further strengthen the proposed IL-6-STAT3-AKT-FoxO1 axis we used genetically modified insulin-insensitive FoxO1$^{ADA}$ mice with a constitutively active form of FoxO1. Inducible FoxO1$^{ADA}$ has mutations within the AKT phosphorylation sites and cannot be phosphorylated by AKT[45]. Induction of FoxO1 using TAT-Cre resulted in constitutively increased and decreased nuclear and cytoplasmatic FoxO1 in bSMCs, respectively (Fig. 5L). Bronchial SMC expressing FoxO1$^{ADA}$ were protected from the IL-6 + R-induced nuclear-to-cytoplasmatic shift (Fig. 5M), indicating an AKT-dependent IL-6-STAT3-mediated inactivation of FoxO1. To further substantiate the IL-6-STAT3-FoxO1 axis in bSMC we isolated bSMC from IL-6$^{-/-}$ at P21 and demonstrated a reduction of phosphorylated STAT3. This finding was linked to a significant increase in nuclear and reduction in cytoplasmatic FoxO1. Finally, bSMCs from IL-6$^{-/-}$ were less viable and proliferation was reduced by 50% (Fig. 5N–P). We next ruled out a possible feedback mechanism from FoxO1 to STAT3. Inhibition of FoxO1 neither affected phosphorylation of STAT3 nor prevented IL-6-mediated STAT3 activation in bSMC (Fig. 5Q). Since the off-target effects of this FoxO1 inhibitor are not extensively described, the observed effects of FoxO1 inhibition were further supported using bSMC from genetically modified mice with SMC-specific ablation of FoxO1 (SMC$^{FoxO1\ Ko}$). While bSMC$^{FoxO1\ Ko}$ exhibited increased viability and proliferation, phosphorylation of STAT3 was not different in bSMC$^{FoxO1\ Ko}$ when compared to SMC$^{WT}$. Despite a significant IL-6-mediated activation of STAT3 in bSMC$^{FoxO1\ Ko}$, we did not determine an effect of IL-6 on viability or proliferation of bSMC$^{FoxO1\ Ko}$ (Fig. 5R, S), suggesting that IL-6-induced proliferation of bSMCs

is FoxO1-dependent. The findings from bSMC$^{WT}$, bSMC with FoxO1$^{ADA}$, bSMC$^{IL-6-/-}$, and bSMC$^{FoxO1\ Ko}$ when coupled with the data after using inhibitors of STAT3 and AKT signaling as well as FoxO1 in bSMC, support the importance of the IL-6-STAT3-AKT-FoxO1 signaling as a regulator of bSMC homeostasis.

**IL-6-FoxO1 signaling is also active in human bSMC and regulates proliferation**. To support our observation in murine bSMC we next investigated primary human bSMCs (hbSMC). First, we confirmed that IL-6 + R activates STAT3 and AKT signaling (Fig. 6A, B). Inhibition of IL-6 signaling using Stattic (a STAT3 inhibitor) blocked AKT signaling, supporting an IL-6-STAT3-AKT axis in hbSMC (Fig. 6B). Second, inhibition of AKT signaling prevented the IL-6 + R-induced nuclear-cytoplasmic shuttling of pFoxO1 in hbSMC (Fig. 6C). Similarly, both STAT3 and AKT inhibition prevented the insulin- and IL-6 + R-induced nuclear-to-cytoplasmic shift of pFoxO1 in hbSMC (Fig. 6D, E). Third, IL-6 + R increased viability of hbSMC, whereas activation of FoxO1 and inhibition of STAT3 using Paclitaxel and Stattic, respectively, reversed this effect. Combining Paclitaxel and STAT3 inhibition did not have an additive effect in hbSMC (Fig. 6F). These findings were further supported by the reduction of nuclear and the increase of cytoplasmic pFoxO1 along with a significant loss of total FoxO1 in hbSMC after exposure to IL-6 + R. In contrast, Paclitaxel had the opposite effect, inducing nuclear sequestration of pFoxO1 and reversing IL-6-mediated inactivation of FoxO1 in hbSMCS (Fig. 6G, H). Finally, IL-6 + R promoted, whereas Paclitaxel blocked and reversed the IL-6-mediated proliferation in hbSMC (Fig. 6I). In summary, our findings in murine and human bSMC indicate an IL-6-induced SMC hyperproliferation through a FoxO1-dependent mechanism, which is reversed in hbSMC by treatment with Paclitaxel.

**IL-6 deficiency prevents bronchial and vascular wall remodeling after perinatal HFD**. After investigating WT mice, we next tested whether the loss of IL-6 protects the lung from the effects of perinatal HFD. The experiments using WT and IL-6$^{-/-}$ mice, both C57BL/6, were performed sequentially and under the same conditions. We exposed female IL-6$^{-/-}$ mice to our metabolic programming model and studied the offspring at P21 and P70. IL-6$^{-/-\ HFD}$ exhibited overweight and an increased relative epigonadal WAT at P21; at P70, however, both groups had similar body weights and relative amounts of epigonadal WAT. The obese phenotype at P21 was related to elevated serum concentration of leptin, but not of insulin, and an impaired glucose tolerance when compared to IL-6$^{-/-SD}$ (Fig. 7A–C).

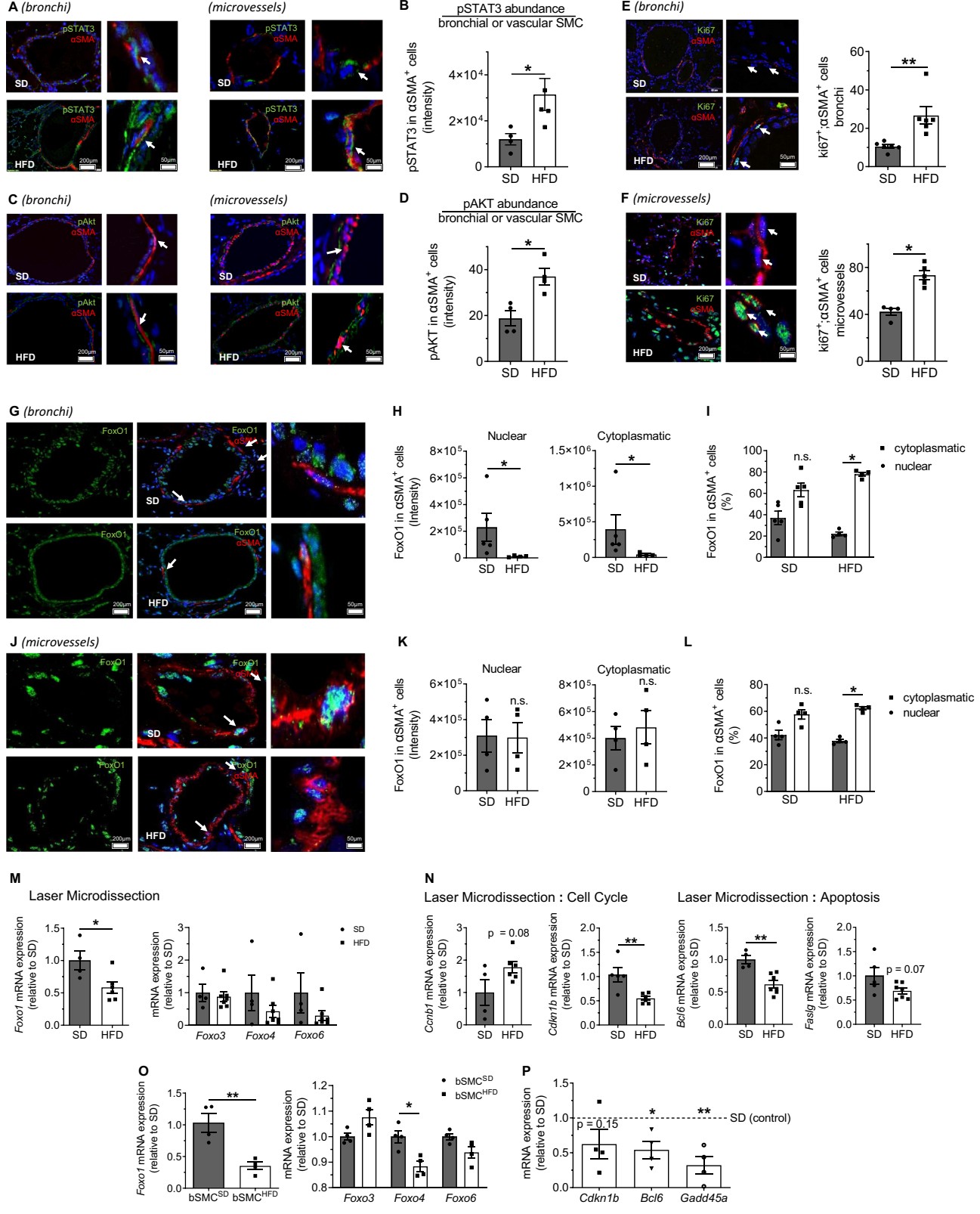

Histomorphometric analysis of epigonadal WAT showed an increase of adipocyte size in IL-6$^{-/-}$ $^{SD}$ when compared to WT$^{SD}$. Prior studies from our group showed a mild adipocyte hypertrophy in epigonadal WAT of WT$^{HFD}$ mice compared to WT;$^{SD}$[46] IL-6$^{-/-}$ protects from this effect. In addition, we did not determine differences in the number of CD68$^+$ cells (macrophages) in WAT between SD and HFD (Supplementary

Fig. 9A–D). Moreover, epigonadal WAT of IL-6$^{-/-}$ $^{HFD}$ showed elevated *lep* as well as unaltered adiponectin and IL-1β mRNA when compared to IL-6$^{-/-}$ $^{SD}$ (Supplementary Fig. 1A, B).

To determine if IL-6 deficiency protects from lung structural changes, we measured the bSML and airway resistance at P70 as described for the WT mice. IL-6$^{-/-}$ $^{HFD}$ exhibited a decrease in bSML and were protected against increased airway resistance

**Fig. 4 A maternal high-fat diet (HFD) induces proliferative STAT3 and AKT signaling and inactivates FoxO1 in bronchial and vascular smooth muscle cells (SMC). A–D** Representative dual immunofluorescent stainings for phosphorylated STAT3 (pSTAT3) or phosphorylated AKT (pAKT) and α-smooth muscle actin (αSMA) of lung sections (bronchi and microvessels) from offspring of HFD- or standard diet (SD)-fed dams at postnatal day 21 (P21) (**A**, **C**). The amount of pSTAT3 and pAKT was quantified and is displayed as the intensity of pSTAT3 or pAKT per αSMA-positive cell (**B**, **D**). **E**, **F** Representative images showing a dual-staining for αSMA and Ki67 (proliferation marker) of the bronchi (**E**) and microvessels (**F**) of offspring at P21; quantification of Ki67-positive nuclei relative to total nuclei among αSMA-positive cells in the bronchi (diameter: 150–250 µm) and microvessels. **G–I** Representative co-immunofluorescence staining to detect αSMA (red), FoxO1 (green), and DAPI (blue, nuclear staining) (**G**). Quantitative assessment of the nuclear and cytoplasmatic FoxO1 intensities (**H**) and the nuclear and cytoplasmic FoxO1 levels relative to total FoxO1 in αSMA-positive bronchial cells (**I**). **J–L** Representative co-immunofluorescence staining of αSMA (red), FoxO1 (green), and DAPI (blue) (**J**). Quantification of the nuclear and cytoplasmatic FoxO1 intensities (**K**), and nuclear and cytoplasmic FoxO1 relative to total FoxO1 in αSMA-positive microvascular SMCs (**L**). **M**, **N** qRT-PCR for gene expression in laser-microdissected bronchi and vessels: *Foxo1*, *Foxo3*, *Foxo4*, and *Fox6* mRNA (**M**) as well as genes regulated by FoxO1: *Ccnb1*, *Cdkn1b*, *Bcl6*, and *Faslg* mRNA (**N**). **O**, **P** qRT-PCR assessment of *Foxo1*, *Foxo3*, *Foxo4*, *Fox6*, *Cdkn1b*, *Bcl6*, and *Gadd45a* mRNA expression in primary bronchial SMCs from HFD and SD offspring (bSMC$^{HFD}$ and bSMC$^{SD}$); SD (control) is set to 1. Data are shown as mean ± standard error of the mean; **B**, **D**, **E**, **F** SD n = 4–6, HFD n = 5–6; **H**, **I**, **K–P** SD n = 4–5 HFD n = 4–7. The statistical analysis was conducted using the two-sided Mann–Whitney test or two-sided Student's *t* test; *P < 0.05; **P < 0.01. Gray = standard diet; white = high-fat diet. A detailed list of sample sizes and *P* value per graph is provided in the Supplemental Material. Source data are provided as a Source Data file.

when compared to IL-6$^{-/-}$ $^{SD}$ (Fig. 7D–F). Similarly, IL-6$^{-/-}$ prevented muscularization of the microvasculature and the increase in medial wall thickness (Fig. 7G–I). Moreover, perinatal HFD did not enhance the expression of *Nppa* in IL-6$^{-/-}$ mice, indicating protection from an increase in right ventricular pressure (Fig. 7J). To investigate whether this protective effect of IL-6 knockout against bronchial and vascular remodeling was related to reduced SMC proliferation, we performed dual immunostaining for Ki67 and αSMA. We observed no increase in proliferation of either bSMC or vascular SMC in IL-6$^{-/-}$ $^{HFD}$ relative to IL-6$^{-/-}$ $^{SD}$ (Fig. 7K, L). Finally, IL-6$^{-/-}$ protected from a nuclear-to-cytoplasmatic shift of FoxO1 in vascular and bronchial SMCs in HFD (Fig. 7M, N). In summary, IL-6 deficiency does not protect against early postnatal obesity but prevents FoxO1 inactivation and bronchial and lung micro-vascular SMC hyperproliferation, thereby preventing bronchial and vascular remodeling after maternal and perinatal obesity.

**SMC-specific FoxO1 ablation causes thickening of the bronchial SML and airway resistance.** To study the functional role of FoxO1 in vivo, we used mice with a specific ablation of FoxO1 in SMC (SMC$^{FoxO1 KO}$) and the respective littermate control, and measured both the bSML and the airway resistance. FoxO1 deletion induced thickening of the bSML, suggesting SMC hyperproliferation (Fig. 8A, B). These structural changes were associated with increased airway resistance in SMC$^{FoxO1-KO}$ relative to littermate control mice (Fig. 8C). Similarly, the medial wall thickness of the microvessels (diameter: 20–70 µm) was significantly greater in SMC$^{FoxO1 KO}$ (Fig. 8D). To exclude an alternative mechanism through compensatory regulation of other FoxOs, we show that SMC-specific ablation of FoxO1 did not affect the expression of genes encoding *Foxo3*, *Foxo4*, and *Foxo6* (Fig. 8E). Moreover, activation of STAT3 and c-Jun as well as the protein abundance of PPARγ were not altered by the deletion of FoxO1 (Fig. 8F–H). In summary, the specific loss of FoxO1 in SMC yields a similar phenotype as that observed in offspring with maternal and perinatal obesity, characterized by bSML thickening, increased airway resistance, and increased medial wall thickness.

**Paclitaxel treatment prevents bronchial and vascular remodeling after perinatal HFD.** Finally, we pursued a pharmacological approach to prevent the effects of perinatal HFD and metabolic disorders on bronchial and vascular remodeling. We treated SD and HFD with Paclitaxel, a FoxO1 activator, via intravenous injection at P50. The experimental design is depicted

in Fig. 9A. First, we measured the body weight, absolute WAT, and relative WAT at P70. Paclitaxel treatment did affect the body composition of HFD, which had similar body weights as SD, but with a significant increase in absolute and relative WAT (Fig. 9B–D). An up to twofold increase in FoxO1 intensity was observed in the αSMA-positive bronchial and vascular SMC of Paclitaxel-treated HFD relative to vehicle-treated HFD (Fig. 9E). Our assessment of bronchial remodeling revealed that vehicle-treated HFD mice exhibited increased airway resistance and bSML thickening, whereas Paclitaxel attenuated these functional and structural changes after perinatal HFD (Fig. 9F, G). Similarly, Paclitaxel treatment protected offspring with perinatal HFD from vascular muscularization and right ventricular hypertrophy (an indicator of pulmonary hypertension) (Fig. 9H, I). These results demonstrate the ability of Paclitaxel to prevent bronchial and vascular remodeling following perinatal HFD.

We then assessed αSMA-positive cell proliferation using Ki67 to determine whether the observed protective effect of Paclitaxel was linked to a blockade of SMC proliferation. Notably, tissues from vehicle-treated HFD contained significantly more proliferating bronchial and vascular αSMA-positive cells compared to vehicle-treated SD. In contrast, similar numbers of proliferating vascular and bronchial αSMA-positive cells were observed in Paclitaxel-treated HFD and SD, thus demonstrating the FoxO1-mediated blockade of SMC proliferation in response to maternal and perinatal obesity (Fig. 9J, K). We next determined if the effects of PAX are related to activation of other pathways or side effects, such as DNA damage or oxidative damage to DNA at P70. Neither perinatal obesity nor treatment with Paclitaxel activated STAT3, JNK, c-Jun or AKT signaling in lungs at P70 (Supplementary Fig. 10A–D). Moreover, we did not detect an increase in γH2AX-positive nuclei or amount of 8-Oxo-dG in lungs after perinatal obesity and/or exposure to Paclitaxel, indicating no differences in DNA damage or oxidative damage to DNA, respectively (Supplementary Fig. 11). Since the quality and quantification of immunofluorescent staining against 8-Oxo-dG can vary, employment of alternative techniques, e.g., mass spectrometry-based measurement of 8-Oxo-dG should be considered for future studies.

In summary, Paclitaxel treatment activated FoxO1 in the lungs, inhibited bronchial and microvascular SMC hyperproliferation, and thereby prevented bronchial and lung vascular remodeling in the offspring after perinatal HFD.

**Perinatal obesity induces hyperproliferation of SMCs across different vascular beds.** Since we propose that the bronchial and vascular remodeling in the lungs after perinatal obesity is

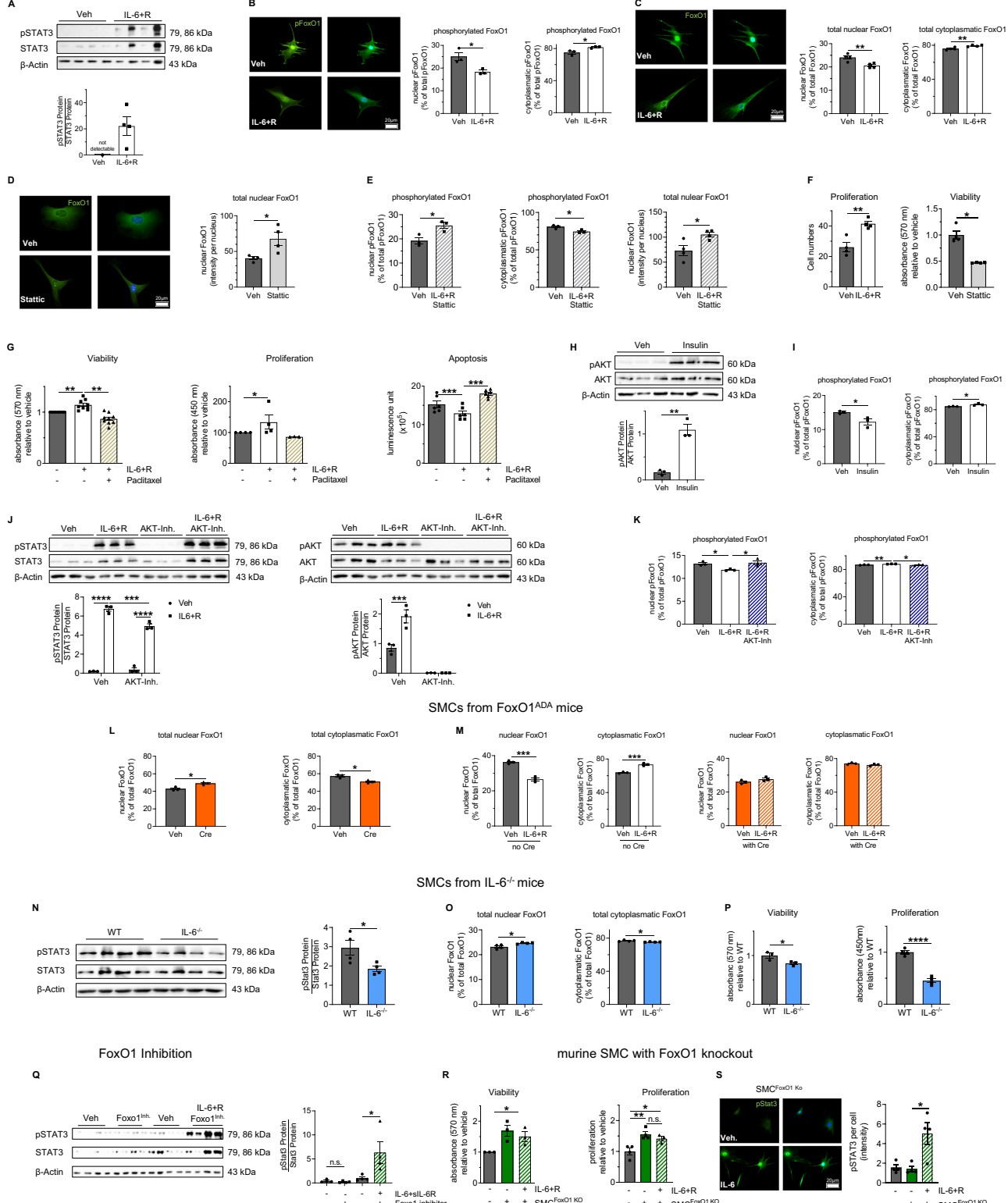

mediated through adipocytokines, we next investigated the systemic effects using kidneys and livers. Histomorphometric analysis revealed an increased vascular SMC layer in kidney and liver in HFD when compared to SD (Supplementary Fig. 12A–D). Complementary cell culture studies using human aortic SMC (haSMC) showed insulin- and IL-6 + R-induced phosphorylation of AKT and STAT3 in haSMC, respectively, and increased

proliferation (Supplementary Fig. 12E–G). Similar to hbSMC, IL-6 + R reduced and increased nuclear and cytoplasmic pFoxO1 in haSMC, respectively, indicating FoxO1 inactivation (Supplementary Fig. 12H). In summary, these additional data demonstrate a systemic IL-6-FoxO1 mechanism by which perinatal obesity induces the proliferation of SMC and could contribute to pulmonary and cardiovascular diseases.

**Fig. 5 Interleukin (IL)−6-STAT3-AKT induces the proliferation of bronchial smooth muscle cells (SMC) via FoxO1. A** Immunoblot showing phosphorylated STAT3 (pSTAT3), and total STAT3 in primary murine bronchial smooth muscle cells (bSMC) from wild-type mice (WT). **B** Nuclear and cytoplasmatic phosphorylated FoxO1 (pFoxO1) fraction in % of total cell pFoxO1 per bSMC; pFoxO1 (green) and DAPI (blue, nucleus). **C** Nuclear and cytoplasmatic total FoxO1 fraction in % of total cell FoxO1 per bSMC. **D** Total FoxO1 in WT-bSMC. **E** Nuclear and cytoplasmatic pFoxO1 fraction in % of total cell pFoxO1 and total nuclear FoxO1 per cell in WT-bSMC. **F** Proliferation (cell count) or viability (MTT assay) in WT-bSMC. **G** Viability (MTT assay), proliferation (BrdU), and apoptosis (active caspase3/7 assay) in WT-bSMC. **H** Immunoblot showing phosphorylated AKT (pAKT), and total AKT in WT-bSMC. **I** Nuclear and cytoplasmatic pFoxO1 fraction in % of total cell pFoxO1 per WT-bSMC. **J** Immunoblots showing pSTAT3, total STAT3, pAKT, and total AKT in WT-bSMC. **K** Nuclear and cytoplasmatic pFoxO1 fraction in % of total cell pFoxO1 per WT-bSMC. **L, M** Nuclear and cytoplasmatic total FoxO1 fraction in % per bSMC from mice with inducible activation of FoxO1 (FoxO1ADA mice). **N** Immunoblot of WT-bSMC and IL-6−/−bSMC showing pSTAT3 and total STAT3. **O** Nuclear and cytoplasmatic total FoxO1 fraction in % of total cell FoxO1 per bSMC. **P** Viability (MTT assay) and proliferation (BrdU). **Q–S** immunoblot for pSTAT3 and STAT3 in WT-bSMC (**Q**). Viability (MTT assay), proliferation (cell count), and pSTAT3 per cell using immunocytochemistry in bSMC from mice with a specific ablation of FoxO1 in SMCs (SMCFoxO1 KO) and WT mice (**R, S**). For immunoblots, β-actin was used as a loading control; densitometric summary is displayed. Interleukin-6 and soluble IL-6 receptor (IL-6 + R), Stattic (STAT3 inhibitor), insulin, AKT Inhibitor (AKT-Inh.), Paclitaxel, and FoxO1 Inhibitor (FoxO1Inh). All bSMC are isolated from 21-day-old mice. Data are shown as mean ± standard error of the mean. **A–S** n = 3–9 biological independent samples or experiments per group; (repeated measure) one-way ANOVA followed by the Bonferroni post-test, two-sided Mann–Whitney test and two-sided Student's; *P < 0.05; **P < 0.01; ***P < 0.001; ****P < 0.0001. Gray = vehicle; white = stimulation; orange=Foxo1;ADA blue=IL-6−/−. A detailed list of sample sizes and P value per graph is provided in the Supplemental Material. Source data are provided as a Source Data file.

### Observational studies

The results of our observational analyses are summarized in Table 1. We observed associations of comparative childhood obesity (i.e., body size at 10 years of age) with lung function (FEV1/FVC) and asthma (β, −0.014; per standard deviation change in FEV1/FVC and childhood obesity). We detected a stronger negative estimation using asthma as outcome (β, −0.188; per standard deviation change in FEV1/FVC and asthma). All association analyses were adjusted for age, sex, region in which the UK Biobank assessment center was located (England, Scotland, and Wales), ethnicity (white, black, Asian, and mixed), and body mass index. A Bonferroni-corrected threshold of 0.025 (adjusting for two comparisons) was used to identify significant associations. To conclude, our analysis shows a negative association of comparative childhood obesity with lung function and asthma.

### Discussion

This study revealed a mechanism by which maternal obesity subverts the physiological growth of the smooth muscle layer in small bronchi and small pulmonary arteries in offspring via IL-6-mediated FoxO1 downregulation. This subversion thus impairs lung function in the offspring and leads to bronchial and vascular remodeling beyond infancy. Our data indicate that FoxO1 activation may be a pharmacological target that would enable reversal of the metabolic programming of bronchial and vascular remodeling by maternal obesity. Here, we provide a molecular concept of the metabolic origin of CLD, based on several major findings. First, maternal obesity-induced early transient obesity with an increase in serum IL-6 concentration and metabolic disorders in offspring. These effects were linked with hyperproliferation of bronchial and vascular SMCs, leading to increased airway resistance, and remodeling of the small bronchi and small pulmonary arteries. Second, FoxO1 was downregulated and sequestered to the cytoplasm in hyperproliferating bronchial and vascular SMC in the lungs of offspring of obese dams. Third, IL-6 enhanced the proliferation, viability, and apoptotic resistance of cultured murine bronchial SMC and human bronchial SMC in a process dependent on STAT3-AKT activity and FoxO1. Fourth, IL-6 deficiency in vivo protected against SMC hyperproliferation, increased airway resistance, and bronchial and vascular remodeling in the offspring of obese dams. Fifth, the genetic ablation of FoxO1 in SMC in vivo reproduced the bronchial and vascular wall remodeling and impaired lung function observed in the offspring of obese dams. Sixth, the pharmacological activation of FoxO1 blocked the hyperproliferation of bronchial and vascular

SMC, inhibited the remodeling of small bronchi and small pulmonary arteries, and prevented the increase in airway and pulmonary vascular resistance in the offspring of obese dams. Finally, we provided evidence of an association of comparative childhood obesity with lung function.

Metabolic programming is the concept wherein adverse nutritive and metabolic influences, e.g., intrauterine growth restriction and early childhood obesity, during a critical window of development are implicated in the early origins of chronic diseases such as obstructive lung disease and PAH[9,13,16,19–21,47]. Our results demonstrate that HFD-induced maternal obesity causes transient early postnatal obesity and metabolic impairment, leading subsequently to hyperproliferative bronchial and vascular remodeling, consistent with the structural changes reported in the lungs of patients with asthma and PAH[48]. Interestingly, patients with overweight or obesity with these diseases experience more severe symptoms, respond less to conventional asthma treatments such as inhaled corticosteroids, and have a poor prognosis[49–52]. These clinical findings, coupled with our observations, suggest that obesity—in this study perinatal obesity being tested—leads to bronchial and vascular remodeling with progressive narrowing and functional impairment. These findings highlight the need for new therapeutic and preventive strategies that target the early origins of lung diseases.

A chronic subacute inflammatory state is a characteristic feature of obesity, linked to the development of metabolic syndrome and CLD pathogenesis[6,7,10,12,22,25,26,28,31,53]. Various inflammatory cytokines and growth factors have been implicated in the abnormal proliferation of lung SMC and the progression of COPD and PAH in this context[2,6,7,25,36]. This inflammatory response is initiated and maintained primarily by WAT-secreted adipocytokines, most notably IL-6[22,24]. We demonstrate here that maternal obesity leads to early-onset obesity with WAT overactivity and elevated serum IL-6 and leptin concentrations in the offspring. Our finding that IL-6−/− were protected from metabolic bronchial and vascular remodeling despite elevated leptin, points out the important role of IL-6. These data are supported by clinical studies demonstrating increased IL-6 concentrations in patients with asthma, COPD, and PAH[26,28,29,53].

Other studies have highlighted the mechanistically important role of IL-6 in animal models and suggested that IL-6 blockade could protect against CLD[2,31]. Similarly, we previously reported a link between early-onset overweight and obesity with activated IL-6/STAT3 signaling, increased airway resistance and lung remodeling[9,20]. Accordingly, prevention of early postnatal

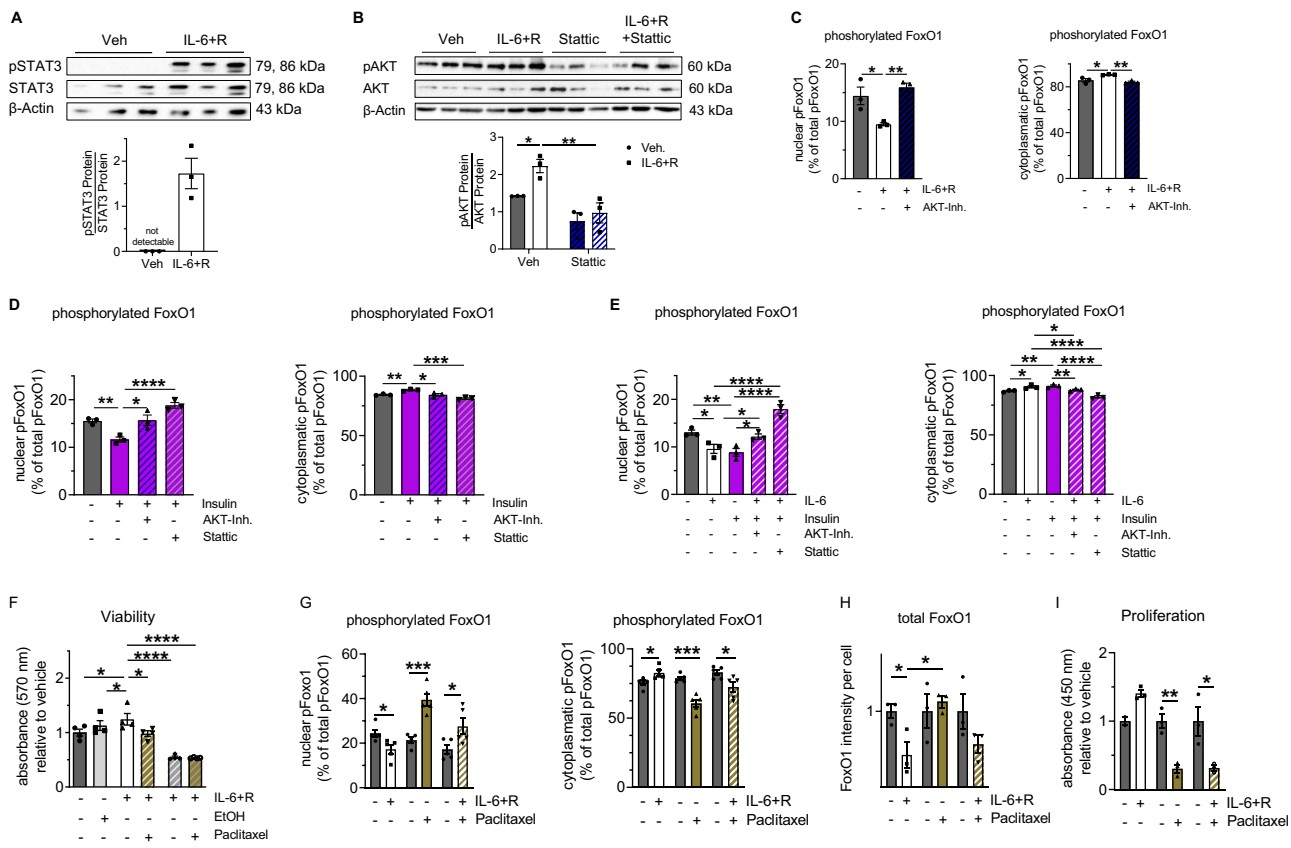

**Fig. 6 Interleukin (IL)−6 regulates the proliferation of human bronchial smooth muscle cells (hbSMC) via FoxO1.** For the experiments with STAT3-inhibitor (Stattic) and AKT inhibitor (AKT-Inh.), hbSMC were pre-exposed to either Stattic, AKT-Inh. or vehicle for 2 h. Subsequently, the hbSMC were stimulated as indicated. **A** Immunoblot showing phosphorylated STAT3 (pSTAT3) in hbSMC after exposure to IL-6 with soluble IL-6 receptor (IL-6 + R) or vehicle for 30 min. **B** Immunoblot showing pAKT and total AKT in hbSMC after exposure to vehicle, IL-6 + R, or Stattic for 30 min. β-actin was used as a loading control. A densitometric summary is displayed. **C** Nuclear and cytoplasmatic phosphorylated FoxO1 (pFoxO1) fraction in % of total cell pFoxO1 per hbSMC using immunofluorescent staining; hbSMC were treated with IL-6 + R alone or combined with AKT-Inh. or vehicle for 30 min. **D** Nuclear and cytoplasmatic pFoxO1 fraction in % of total cell pFoxO1 per hbSMC using immunofluorescent staining; hbSMC were treated with insulin alone or combined with AKT-Inh. or Stattic. **E** Nuclear and cytoplasmatic pFoxO1 fraction in % of total cell pFoxO1 per hbSMC using immunofluorescent staining; hbSMC were treated with IL-6 + R or insulin alone or both in combination with AKT-Inh. or Stattic for 30 min. **F** Viability (MTT assay) of hbSMC after exposure to vehicle, IL-6 + R, Paclitaxel and/or Stattic. **G-I** hbSMCs were exposed to IL-6 + R or vehicle with or without Paclitaxel for 30 min (**G**) or 24 h (**H**, **I**). Nuclear and cytoplasmatic pFoxO1 fraction in % of total cell pFoxO1 per bSMC (**G**); total FoxO1 per hbSMC (**H**), and proliferation (BrdU) of hbSMC (**I**). Data are shown as mean ± standard error of the mean. **A-I** n = 3–5 biological independent samples or experiments per group; (repeated measure) one-way ANOVA followed by the Bonferroni post-test or the two-sided Mann–Whitney test; *P < 0.05; **P < 0.01; ***P < 0.001; ****P < 0.0001. Gray = vehicle; white = stimulation; orange = Paclitaxel; blue = IL-6. A detailed list of sample sizes and P value per graph is provided in the Supplemental Material. Source data are provided as a Source Data file.

overweight protected against these structural and functional changes in the lung[54]. In the present study, IL-6 knockout did not have a beneficial effect on postnatal weight gain, increased serum leptin, and impaired glucose metabolism after maternal obesity, but intriguingly prevented lung SMC hyperproliferation, bronchial and vascular remodeling, increased airway resistance, and elevated expression of the gene encoding natriuretic peptide A (ANP; *Nppa*) as an indicator of higher right ventricular pressure. In cultured bSMCs, an increase in the viability, proliferation, and apoptosis resistance of bSMC after exposure to IL-6 was disclosed as underlying mechanisms, linked to nuclear exclusion and cytoplasmatic sequestration of FoxO1 via STAT3 signaling. Moreover, our data indicate that IL-6-STAT3 mediated phosphorylation of FoxO1 is AKT-dependent. Given the fact that offspring of obese dams exhibit hyperinsulinemia at P21[42], insulin and IL-6 signaling might converge on FoxO1. In contrast, the Paclitaxel-mediated nuclear translocation of FoxO1 reversed these IL-6-mediated effects, thus further confirming the role of an

IL-6-FoxO1 signaling in the lung SMC. In vivo, dual immuno-fluorescence staining for FoxO1 and αSMA demonstrated a nuclear-to-cytoplasmatic shift of FoxO1 in SMC, which was associated with increased numbers of proliferating bronchial and vascular SMC in the lungs of HFD relative to SD offspring. The loss of FoxO1 expression and transcriptional activity in SMCs explains various key findings associated with lung metabolic programming, including the reduced expression of gene encoding cyclins (e.g., *Ccnb1* (cyclin B1)), increased expression of *Cdknb1b* (p27), and reductions in the expression of pro-apoptotic genes, such as *Bcl6* (B cell lymphoma 6 protein) and *Faslg* (CD95, Fas ligand). These findings may well explain the hyperproliferation of SMC in the lungs of obese dams and in cultures exposed to IL-6/ IL-6R. In addition, initial investigations showed thickening of the vascular SMC layer in kidneys and livers after perinatal obesity as well as increased proliferation and nuclear-to cytoplasmatic shift of FoxO1 in human aortic SMC after treatment with IL-6/sIL-6R. These data indicate a possible systemic IL-6-FoxO1 mechanism

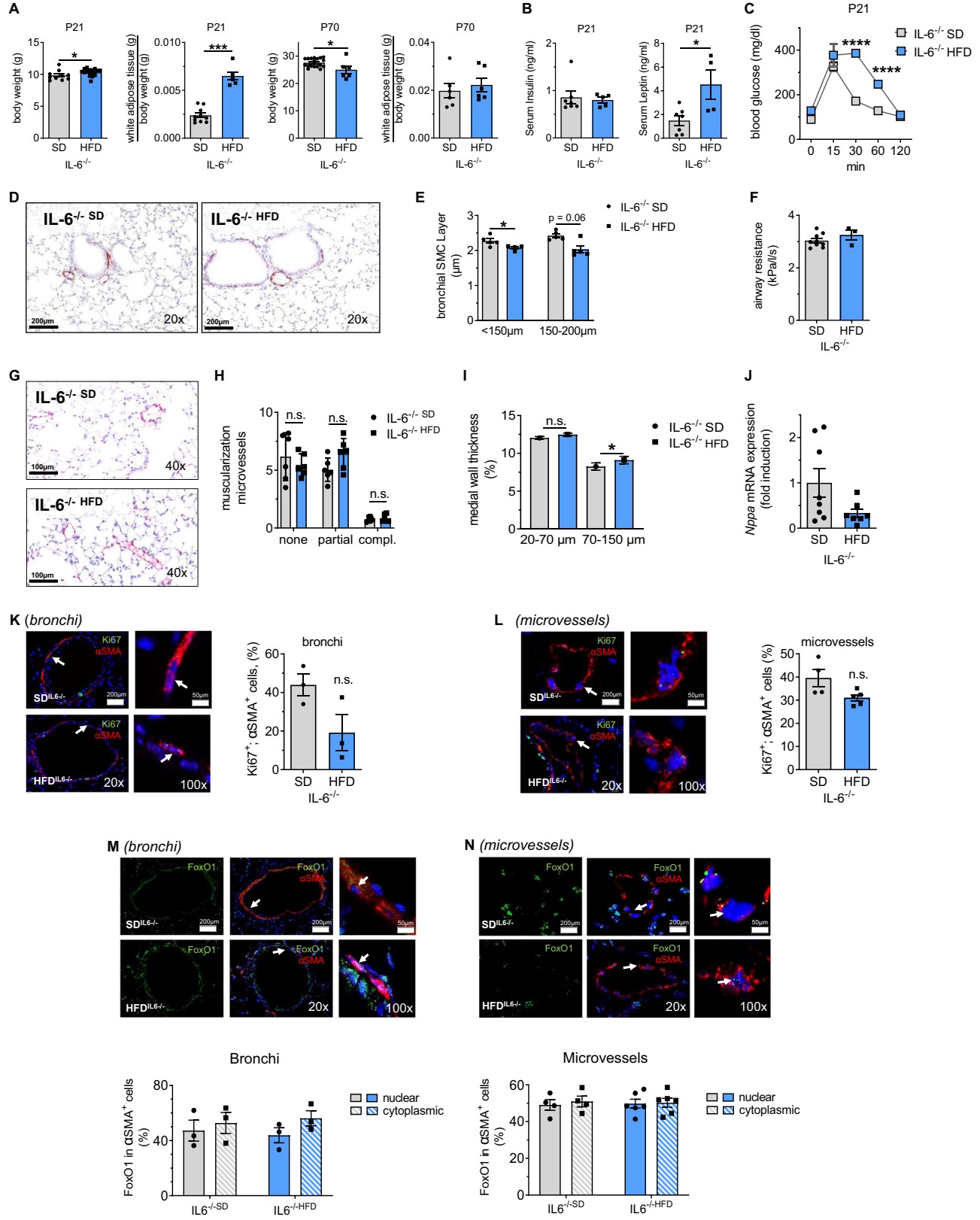

after perinatal obesity that might contribute to the pathogenesis of cardiovascular diseases.

Cell homeostasis relies on the signaling pathways of various growth factors and cytokines, many of them converging at and acting via FoxOs. Recent studies suggested that FoxO family members play crucial roles in the pathogenesis of adult CLDs

such as pulmonary fibrosis and PAH[34]. Consistent with these reports, we demonstrated an IL-6-STAT3-AKT-FoxO1 signaling in bSMC in vitro. These findings were further supported by in vivo studies showing (1) that IL-6 deficiency protected against SMC hyperproliferation and bronchial and vascular remodeling and (2) that SMC-specific ablation of FoxO1 yielded a lung

**Fig. 7 Offspring of Interleukin (IL)−6-deficient dams fed a perinatal high-fat diet (HFD) are protected from structural and functional lung changes at postnatal day 70 (P70) despite early-onset obesity and impaired glucose tolerance. A** Body weight and ratio of white adipose tissue (WAT) relative to body weight of the offspring of HFD- and standard diet (SD)-fed dams at P21 and P70. **B** Serum insulin and leptin concentrations at P21 using ELISA. **C** Offspring of HFD- and SD-fed dams were subjected to an intraperitoneal glucose tolerance test (ipGTT) at P21. **D–F** Representative images of lungs from offspring at P70. Tissues were stained for α-smooth muscle actin (αSMA) to assess the bronchial SMC layer (**D**). Quantitative measurement of the bronchial SMC (bSMC) layer in bronchi with a diameter of <150 μm and 150–200 μm (**E**). Assessment of respiratory airway resistance using body plethysmography at P70 (**F**). **G–I** Representative images of lungs from offspring at P70. Tissues were co-stained for αSMA and von Willebrand factor (vWF) (**G**) to assess the muscularization (**H**) and medial wall thickness of the microvessels (**I**). **J** assessment of *Nppa* mRNA in the right ventricle at P70 using qRT-PCR. **K, L** Representative immunofluorescent images of lungs from offspring at P21. Tissues were stained for αSMA and Ki67. The Ki67-positive nuclei/total nuclei were calculated for αSMA-positive cells in bronchi (diameter: 150–250 μm diameter) (**K**) and in microvessels (20–100 μm) (**L**). Values are shown next to the respective images. **M, N** Representative co-immunofluorescence staining to detect αSMA (red) and FoxO1 (green) in bronchi (**M**) and microvessels (**N**); DAPI was used for nuclear staining. Quantitative assessment of the nuclear and cytoplasmatic FoxO1 fraction in % of total cell FoxO1 in αSMA-positive bronchial (**M**) and vascular cells (**N**). Data are shown as mean ± standard error of the mean. **A–N** SD *n* = 3–19, HFD *n* = 3–17. Data were analyzed using the two-sided Mann–Whitney test and two-way ANOVA followed by the Bonferroni post-test; *$*P < 0.05$; ***$P < 0.001$; ****$P < 0.0001$. Gray = standard diet; white = high-fat diet; blue=IL-6$^{-/-}$. A detailed list of sample sizes and *P* value per graph is provided in the Supplemental Material. Source data are provided as a Source Data file.

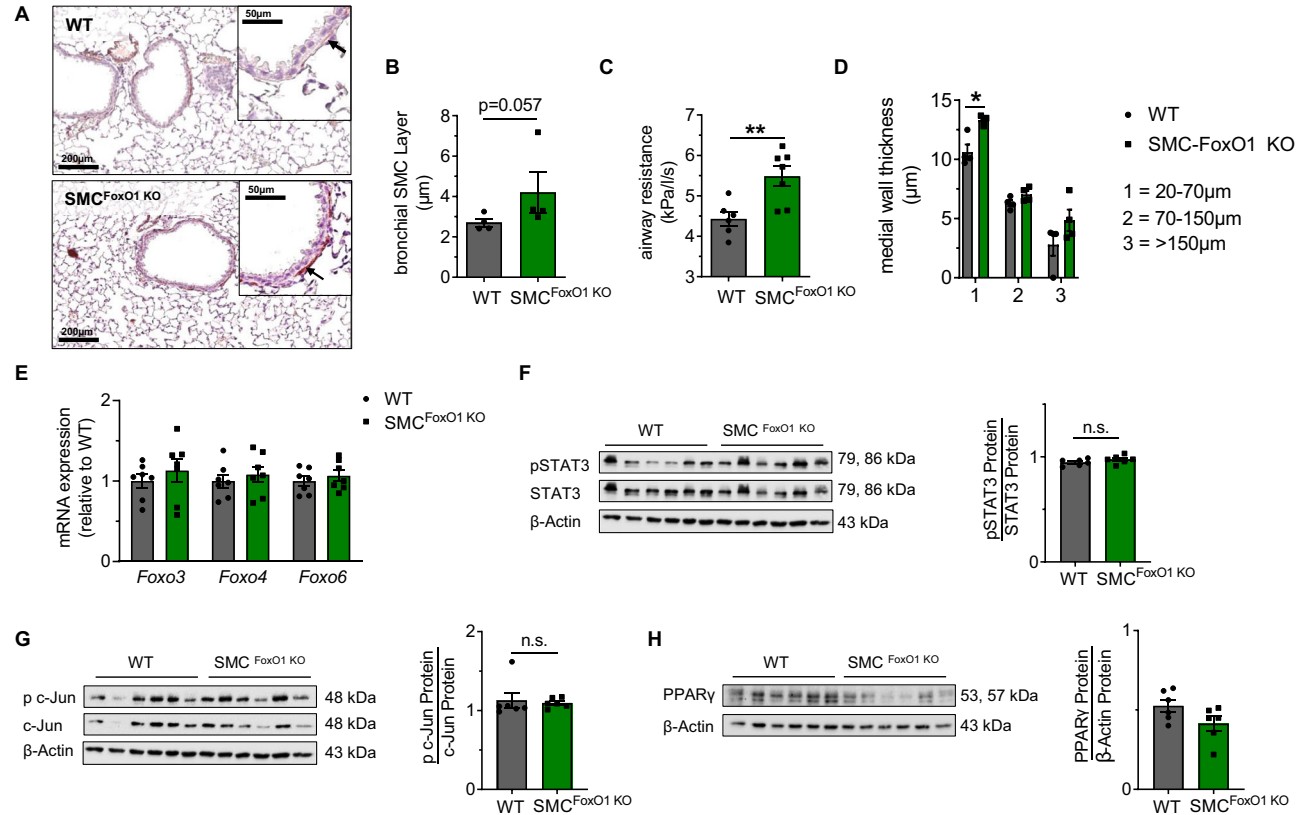

**Fig. 8 Smooth muscle cell (SMC)-specific ablation of FoxO1 induces airway resistance and vascular muscularization in mice.** Genetically modified mice exhibiting a specific ablation of FoxO1 in SMC (SMC$^{FoxO1-KO}$) were compared to the respective wild-type controls (WT). **A** Representative images for α-smooth muscle actin (αSMA), an indicator of SMC, to assess the bronchial SMC layer. **B** Quantitative measurement of the SMC layer in the bronchi (diameter: 150–200 μm) of adult mice. **C** Assessment of respiratory airway resistance in adult mice using body plethysmography. **D** Measurement of the medial wall thickness in pulmonary microvessels with diameters of 20–70 μm, 70–150 μm and >150 μm. **E** Measurement of gene expression of *Foxo3*, *Foxo4*, and *Foxo6* in total lung homogenates using qRT-PCR at P70; β-actin served as housekeeping gene; the expression is shown relative to WT and WT is set 1. **F–H** Immunoblots for pSTAT3 and total STAT3 (**F**), p c-Jun and total c-Jun (**G**), and PPARγ (**H**) in total lung homogenates of WT and SMC$^{FoxO1 KO}$; β-actin served as a loading control. The densitometric summaries of pSTAT3 relative to total STAT3, p c-Jun relative to total c-Jun, and PPARγ relative to β-actin are displayed next to the respective immunoblot. Data are shown as mean ± standard error of the mean. **B–H** WT *n* = 4–6, SMC$^{FoxO1-KO}$ *n* = 4–7. Data were analyzed using the two-sided Mann–Whitney test; *$*P < 0.05$; **$P < 0.01$. Gray = wild-type; orange = Foxo1$^{ADA}$. A detailed list of sample sizes and *P* value per graph is provided in the Supplemental Material. Source data are provided as a Source Data file.

phenotype corresponding to that after maternal obesity. Moreover, both our in vitro and in vivo results demonstrate that in the offspring of obese dams Paclitaxel upregulated nuclear retention of FoxO1 in SMC, and this effect correlated with the inhibition of SMC hyperproliferation in the offspring of obese dams and the reversal of bronchial and vascular SML thickening. We note that the increase in the airway resistance induced by Paclitaxel in SD could be related to bronchial hyperresponsiveness. Our results suggest a new preventive strategy for the early metabolic origins of CLD.

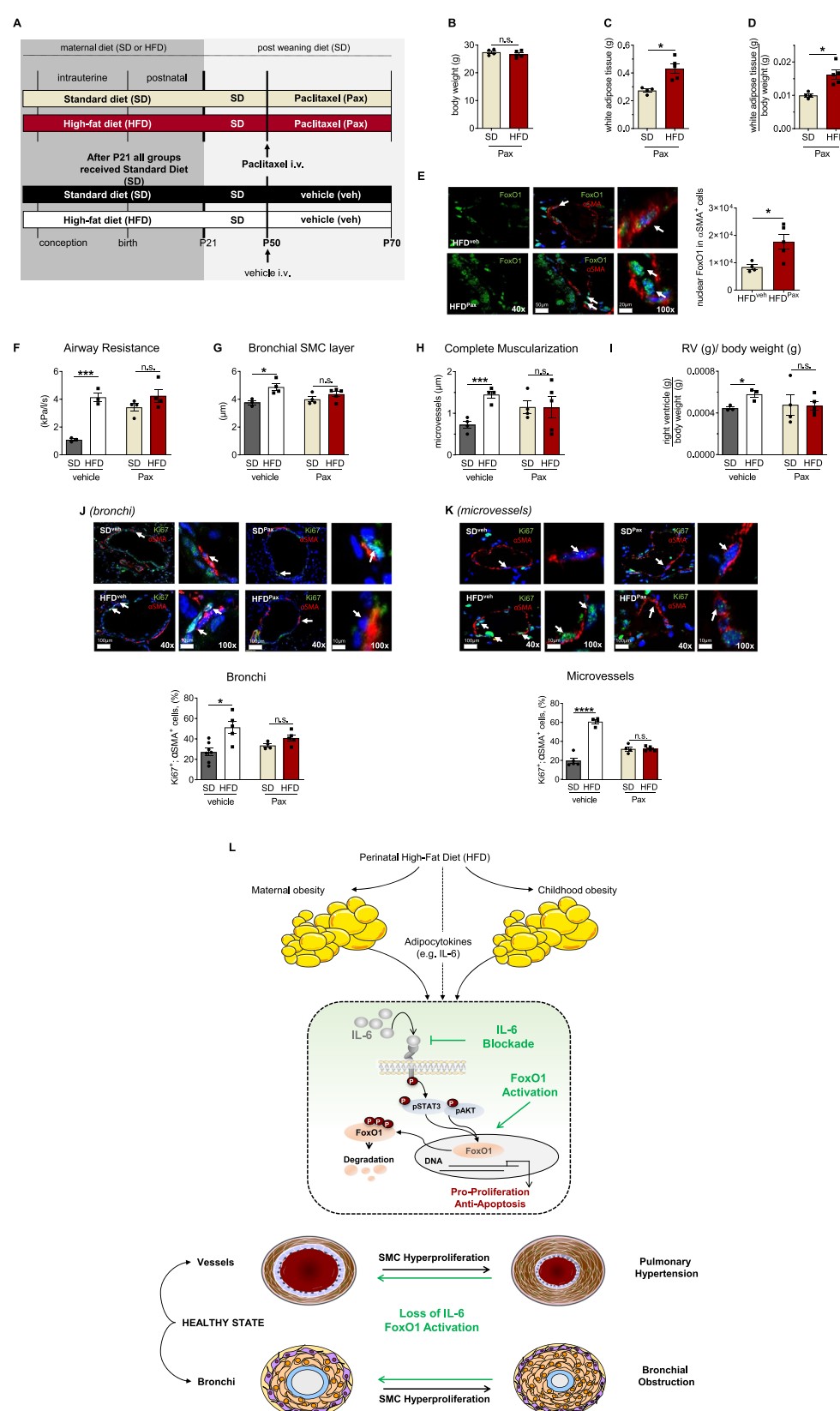

Paclitaxel has been used widely as a cancer chemotherapeutic drug[55]. In this study, Paclitaxel was effective at concentrations up to tenfold lower than those used for anticancer treatment. Therefore, we did not observe previously mentioned side effects, indicating that this dose is below the known toxicity levels. Our observation is consistent with previous studies in which the non-oncological administration of Paclitaxel was used to prevent and/or treat vascular remodeling and neointima formation after coronary angioplasty or in animal models of PAH[56]. One possible mechanism how Paclitaxel regulates FoxO1 could be microtubule stabilization[57].

**Fig. 9 Paclitaxel (Pax) mitigates perinatal high-fat diet (HFD)-associated bronchial and vascular remodeling. A** The offspring of HFD- or standard diet (SD)-fed dams received an intravenous (i.v.) administration of vehicle or Pax, a FoxO1 activator, at postnatal day 50 (P50). All endpoints were assessed at P70. **B–D** Body weight (**B**), white adipose tissue (WAT) (**C**), and WAT relative to body weight (**D**). **E** Representative immunofluorescent lung images for α-smooth muscle actin (αSMA, red) and FoxO1 (green); the quantitative FoxO1 intensities per defined area (10 μm$^2$) are displayed. **F** Respiratory airway resistance. **G** Analysis of the bronchial SMC (bSMC) layer in bronchi (diameter: 150–250 μm). **H, I** Complete microvascular muscularization (20–100 μm) (**H**). Right ventricular weight relative to body weight (**I**). **J, K** Representative immunofluorescent lung images for αSMA (red) and Ki67 (green, proliferation); the number of Ki67$^+$ nuclei relative to total nuclei of αSMA-positive cells in bronchi (diameter: 150–250 μm) (**J**) and microvessels (20–100 μm) (**K**). Values are shown under the respective images. Data are shown as mean ± standard error of the mean. **B–K** SD $n = 3–7$, HFD $n = 3–5$; two-sided Mann–Whitney test and two-sided Student's $t$ test; *$P < 0.05$; ***$P < 0.001$; ****$P < 0.0001$. Black = SD + vehicle; white = HFD + vehicle; beige = SD + Paclitaxel; red = HFD + Paclitaxel. A detailed list of sample sizes and $P$ value per graph is provided in the Supplemental Material. Source data are provided as a Source Data file. **L** Proposed working model: effects of HFD-induced maternal and perinatal obesity with early-onset offspring obesity and elevated circulating levels of the adipocytokine Interleukin (IL)-6 on the developing lung. During lung development, this chronic subacute inflammatory state promotes the activation of STAT3 and AKT signaling in the lung and the proliferation of smooth muscle cells (SMC) via the cytoplasmatic sequestration of the transcription factor FoxO1. Bronchial and vascular SMC hyperproliferation is ultimately related to bronchial obstruction and pulmonary hypertension, respectively. Both IL-6 deficiency and FoxO1 activation prevent these structural and functional impairments of the bronchi and vessels after perinatal obesity, suggesting an IL-6-STAT3-AKT-FoxO1 axis and a new therapeutic approach to mitigate the early metabolic origins of chronic lung diseases (CLDs). The illustration has been created by the authors.

**Table 1 Observational associations of expiratory volume in 1 s/forced vital capacity (FEV1/FVC) ratio Z-score and asthma with comparative body size at 10 years of age.**

|  | Beta | SE | P |
|---|---|---|---|
| FEV1/FVC | −0.014 | 0.005 | 2.98E-03 |
| Asthma | −0.188 | 0.056 | 7.51E-04 |

Thus, Paclitaxel or derivatives thereof may offer a potential therapeutic route for the reversal of the bronchial and vascular remodeling induced by adverse perinatal metabolic influences and could potentially mitigate the risk of CLD beyond infancy.

Our observational findings indicated an association of childhood obesity with reduced lung function. However, these data refer to childhood obesity and can only be used as a surrogate for maternal or perinatal obesity. Moreover, consistent with our findings, previous Mendelian randomization analyses indicated a causal effect between birth weight and reduced lung function in adulthood[58]. In addition, previous experimental and clinical studies identified metabolic disorders such as hyperinsulinemia as risk factors for PAH[59,60]. Clinical data indicate that overweight and obesity aggravate the courses of CLDs including asthma and COPD[2,10,51]. Our data further confirm this association by demonstrating associations of childhood overweight with impaired lung function and asthma, thereby documenting the urgent need to define preventive and therapeutic strategies for overweight children at risk of CLD.

Obesity is becoming a global epidemic in which various pathologies are mediated by a chronic subacute inflammatory state. Exposure to adverse conditions such as maternal obesity during a critical period of development induces life-long structural and functional changes in the lung. The current lack of effective preventive and therapeutic measures underscores the need to identify targets for risk mitigation and treatment in this population. Our observation that perinatal maternal obesity mediates bronchial and vascular SMC hyperproliferation via the IL-6-mediated nuclear exclusion of FoxO1 provides a potential target for such preventive and/or therapeutic interventions (Fig. 9).

## Methods

**Animal procedures.** The experiments were performed in accordance with German regulations and legal requirements and were approved by the local government authorities (LANUV, NRW, Germany; AZ # 2018A320, AZ # 2012A424, AZ # 2011-025, and AZ# 50.15.015; Regierungspräsidium Gießen und Darmstadt, Hessen, Germany; approval numbers B2/318; B2/311). All mice were housed in a room at 22 ± 2 °C and exposed to a 12-h light/dark cycle. Mice were kept in Type II long individually ventilated cages (Tecniplast, Italy) under standard husbandry conditions according to the Directive 2010/63/EU and were free from FELASA-listed infectious agents.

**Animal model of metabolic programming.** C57BL/6 (WT) and B6.129S2IL-6tm1Kopf/J (IL-6$^{−/−}$) were studied. Virgin female WT and IL-6$^{−/−}$ mice from our own colony served as future dams. After weaning [postnatal day 21 (P21)], they either received a high-fat diet (HFD;$^{mat}$ modified #C1057, Altromin, Lage, Germany) for induction of obesity or a standard diet (SD, SD;$^{mat}$ ssniff #R/M-H, V1534-0) for 7 weeks as previously described[42]. At the end of the 7 weeks of HFD or SD feeding, female mice received an intraperitoneal (i.p.) glucose tolerance test (GTT). Subsequently, HFD$^{mat}$ and SD$^{mat}$ females were time-mated with SD-fed males, checked for vaginal plugs the following morning, and continued on their respective diet. After birth, the litter size of all dams was normalized to six for each litter. Water and chow were available ad libitum and food was withdrawn only for experimental reasons. After weaning at P21, male offspring of both groups were fed SD until P70, defining two groups as shown in Fig. 1A: (1) SD (SD$^{mat}$ followed by SD during gestation and lactation), and (2) HFD (HFD$^{mat}$ followed by HFD during gestation and lactation; also referred to as maternal or perinatal obesity). The exact numbers of animals are listed in the figure legends and Online Supplement. To avoid gender-specific differences, only male mice were included in the experiments.

**Genetically modified mice with specific ablation of Foxo1 in SMC.** Mice with a smooth muscle cell-specific deletion of *Foxo1* were generated by crossing mice homozygous for the floxed *Foxo1* allele with *Sm22-cre* mice (*Tagln-cre* 1Her/J, the Jackson Laboratory, Bar Harbor, ME, USA)[34]. The offspring, *Foxo1*$^{flox/flox}$-Sm22-cre (SMC$^{Foxo1-KO}$), and their littermates, control *Foxo1*$^{+/+}$-Sm22-cre (WT), were used for experiments. Ten- to 12-week-old male SMC$^{Foxo1-KO}$ and their controls were analyzed. FoxO$^{ADA-fl/fl}$ were kindly provided by Dr. Thomas Wunderlich (Max Planck Institute for Metabolic Research, University Hospital Cologne, Cologne, Germany).

**Genotyping.** IL-6$^{−/−}$ and mice with loxP-site targeted alleles were genotyped by PCR. For genotyping mouse IL-6$^{−/−}$ lines, the 5′ and 3′ primers were oIMR0212 5′-TTCCATCCAGTTGCCTTCTTGG-3′ and oIMR0213 5′-TTCTCATTTC-CACGATTTCCCAG-3′ and oIMR0214 5′-CCGGAGAACCTGCGTGCAATCC-3′. DNA was extracted from mouse tail biopsies and 2 μl (5–20 ng) were used in the subsequent PCR reaction. Cycling conditions were 3 min at 94 °C, 35 cycles of 30 s at 94 °C, 1 min at 60 °C, 1 min at 72 °C followed by 10 min at 72 °C. The WT alleles resulted in 174-bp bands, the mutant in 380-bp bands and the heterozygous in both 380-bp and 174-bp bands. For genotyping Foxo1 lines, the 5′ and 3′ primers were CK30 5′-GTGTACAAACCAGCTGAGCAC-3′; CK142 5′-CCTTCAGAGCTGCCAGGTGAATATG-3′ and CK143 5′-TGTAGAGAGTA TGCCGTCAGAGTGAG-3′. DNA was extracted from mouse tail biopsies, and 2 μl (5–20 ng) were used in the subsequent PCR reaction. Cycling conditions were 5 min at 95 °C; 35 cycles of 30 s at 95 °C, 20 s at 64 °C and 60 s at 72 °C, followed by 7 min at 72 °C. The WT and F2-3p (floxed) alleles resulted in 290-bp and 350-bp bands, respectively[34].

**Treatment with Paclitaxel.** To determine if activation of FoxO1 protects offspring of obese dams (HFD) from bronchial and vascular remodeling, offspring of both groups HFD and SD received a single intravenous (tail vein) injection of Paclitaxel [NeoTaxan (6 mg/ml), Hexal, Hochzkirchen, Germany] at P50; control mice were treated with 0.9% NaCl. At P70, the animals underwent a lung function test to assess airway resistance and lungs were excised for assessment of bronchial and

vascular remodeling as well as the proliferation of bronchial and vascular SMCs. Right ventricular weight (g) was obtained and related to body weight (g).

**Physiological data of dams and offspring**. The body weight [BW; gram (g)] of the future dams was measured at weeks 1, 3, 5, and 7 after starting HFD or SD as well as at gestational day 1 (G1) and at weaning (P21). At P21 and P70, we obtained BW and the amount of epigonadal white adipose tissue (WAT) of the offspring in g.

**Measurement of airway resistance (Res) and bronchial hyperreagibility**. Respiratory system resistance (Res) was assessed using direct plethysmography for mice (FinePointe™RC; Buxco, Wellington, NC, USA). To this end, mice were deeply anesthetized via ip injection of Xylazine (10 mg/kg BW; Rompun®, Bayer Vital GmbH, Leverkusen, Germany) and Ketamine (100 mg/kg BW; Ketaset®, Zoetis, Berlin, Germany), and a tracheotomy was performed. The trachea was dissected, a surgical suture was placed between the trachea and esophagus, the trachea was incised between tracheal cartilages, a tracheal tube was inserted, and the tube was fixed by a ligature to avoid any leak in the system. Next, the tracheal tube was connected to the tracheal manifold of the plethysmograph, and an esophageal probe was inserted. After acclimatization for 5 min, Res was measured as previously described[9]. To test for bronchial hyperreagibility, we measured Res at baseline after nebulization with PBS, followed by stimulation with methacholine—a bronchoconstrictor—in four different doses (2.5, 6.25, 12.5, and 25 mg/ml) and measurement of Res as described[9].

**Echocardiography**. To determine heart rate (HR), cardiac output (CO), stroke volume (SV), right ventricular internal diameter (RVID), tricuspid annular plane systolic excursion (TAPSE), and left ventricular (LV) ejection fraction (EF) transthoracic echocardiography was performed as described previously[61–64]. In short, the isoflurane anesthesia was applied to the mice by keeping them in an anesthesia induction chamber with gas flow adjusted to 2–3 l per min. A vaporizer (VisualSonics, Toronto, Canada) was used to deliver isoflurane at 3%. After induction, anesthesia was sustained on 1–1.5% isoflurane in room air supplemented with 100% $O_2$. Mice were fixed in a supine position on a homeothermic plate with all the legs attached to electrocardiogram electrodes to supervise the heart rate. To monitor body temperature a rectal thermometer (Indus Instruments, Houston, TX, USA) was used. Prior to measurement, the chest was shaved and a chemical hair remover was applied in addition to reduce ultrasound attenuation. Subsequently, an ultrasonic gel was applied to the chest wall of the mouse. The transthoracic echocardiography was conducted with the VEVO770 high-resolution imaging system equipped with a 25-MHz transducer (RVM710B) or with the VEVO2100 system equipped with an 18- to 38-MHz (MS400) transducer (VisualSonics). RVID was determined as the maximal distance between the septum and the right ventricle-free wall using the apical four-chamber view. Also in the apical four-chamber view, the TAPSE was measured, by orienting the M-mode cursor to the junction of the tricuspid valve plane and the right ventricle-free wall.

**Hemodynamics**. Hemodynamic properties were analyzed as described[65]. Briefly, anesthesia was given to mice as described above. The mice were intubated and fixed in a supine position on a thermic plate (AD Instruments, Spechbach, Germany) and they were ventilated with a small-animal ventilator (MiniVent type 845, Hugo Sachs Elektronik, March-Hugstetten, Germany). Body temperature was measured by a rectal probe (AD Instruments) and was maintained at 37 °C. Subsequently, mice were catheterized on the right external jugular vein with a high fidelity 1.4 F micromanometer/Mikro-Tip Pressure catheter (Millar Instruments, Houston, TX, USA). The catheter was then advanced into the right ventricle to obtain the RVSP. Next, the systolic blood pressure (SBP) and the diastolic blood pressure (DBP) were measured by guiding the 1.4 F micromanometer catheter into the aorta and the LV through the left carotid artery. Data collection and analysis were performed with the PowerLab data acquisition system (MPVS-Ultra Single Segment Foundation System, AD Instruments) and LabChart 7 for Windows software. Pulmonary vascular resistance index (PVRI) was calculated using the following formula: PVRI = RVSP/CI, where RVSP is right ventricular systolic pressure (mm Hg). In addition, we measured left ventricular pressure (LVP).

**Cardiac preparation**. Tissue preparation. After exsanguination, the left lung was fixed for histology in 10% neutral buffered formalin, and the right lung was snap-frozen in liquid nitrogen. For right-heart hypertrophy, the right ventricle (RV) was separated from the left ventricle plus septum (LV + S), and after weighing in the tissue the RV/BW ratio was determined.

**Measurement of serum adipocytokine concentrations using ELISA**. Measurements of murine serum adiponectin and IL-6 levels were performed using commercially available ELISA kits according to the manufacturer's instructions: Mouse adiponectin ELISA (BioVendor, #RD293023100R, Heidelberg, Germany) and Mouse IL-6 ELISA (R&D Systems, #M6000B, Minneapolis, Minnesota, USA). Murine serum leptin levels were measured in duplicates in a multiplex analyzer (Bio-Plex 200®, Bio-Rad Laboratories, USA) according to the manufacturer's

instructions (Milliplex® MAP, MA, USA). Sera were thawed, centrifuged for 5 min at 10,000×g and 4 °C, and were added in a dilution 1:4 to the appropriate wells. By using the median fluorescence intensity and the standard curve, the absolute concentration of each cytokine (pg/ml) was calculated (Bio-Plex Manager 6.1, Bio-Rad Laboratories).

**Intraperitoneal glucose tolerance test**. Dams ($SD^{mat}$ and $HFD^{mat}$) and offspring underwent an intraperitoneal glucose tolerance test (i.p. GTT) prior to mating and at P21 as well as P70, as previously described[66]. Animals were fasted overnight (12 h) while having unlimited access to water. After 12 h of fasting, blood glucose levels were determined by tail vein blood sample collection, followed by i.p. injection of 20% glucose (10 ml/kg BW). Blood glucose levels were measured using a glucometer (GlucoMen® LX, A.Menarini diagnostics, Berlin, Germany) after 15, 30, 60, and 120 min.

**Laser-microdissection**. Microdissection of bronchi, lung vessels, and adjacent tissue was performed with PFA-fixed and paraffin-embedded lung tissue sections (13 µm) on MembraneSlides (Leica #11600288, Wetzlar, Germany). Tissues were deparaffinized in isopropanol (Roth, #6752, Karlsruhe, Germany) and rehydrated in a graded ethanol series (100%, 96%, 80%, and 70% for one minute each) to DEPC-$H_2O$. Sections of bronchi and vessels were dissected with PALMProbe software (Zeiss, Oberkochen, Germany) at PALMMicroBeam microscope (Zeiss) through laser-microdissection with laser settings at speed 73, energy 56 µJ, and focus 88. Laser-microdissected bronchi and vessels were collected in AdhesiveCap 500 opaque tubes (Zeiss, #415190-9201-000). After collecting the dissectates, samples were incubated with protein kinase K (Fermentas, Thermo Fisher Scientific, #EO0491, Waltham, Massachusetts, USA) overnight to dissolve the lung tissue out of the adhesive cap. Afterward, RNA was extracted according to an established protocol with TRI reagent (Sigma-Aldrich, #T9424-200ml, St. Louis, MO, USA).

**RNA-Seq analysis**. Raw fastq files were trimmed using BBduk[67] and mapped to the Mus musculus genome (GhCm38, using gene annotation v97) with the STAR aligner[68] according to Lexogene recommendations. Sequence reads were assigned to genomic features using feature counts[69] and DeSeq2[70] was used for statistical evaluation of significant differentially expressed genes between SD and HFD mice with a q-value cut-off of 0.05. Plots were created using pheatmap and heatmaply[71] and manhattanly for interactive volcano plots, for gene ontology enrichment analysis, and pathway discovery we used clusterProfiler[72], gene annotation was performed with BioMart[73]. Promotors were defined based on proximity with the transcription start site (TSS) described for the genes. For the analysis, 2000-bp region upstream of the TSS was selected. Distal enhancers were not included.

**Tissue preparation**. The mice were deeply anesthetized via i.p. of Xylazine (10 mg/kg BW) and Ketamine (100 mg/kg BW). After exsanguination by cardiac puncture and bleeding, the lung and the trachea were dissected. First, the right main bronchus was ligated with a surgical thread, the right lung was excised and immediately snap-frozen and stored at −80 °C. Second, an incision of the trachea between tracheal cartilages and intubation with a cannula (26 G) was performed, followed by fixation using a surgical thread placed between the trachea and the esophagus. Subsequently, lungs were fixed with 4% paraformaldehyde (PFA) at a pressure of 20 cm $H_2O$ for paraffin embedding and sectioning. Once the abdominal wall was opened, the total amount of the epigonadal WAT, the heart, the kidney and the liver were excised and weighed. A part of the WAT was immediately snap-frozen, and stored at −80 °C for further analyses; the rest of the WAT was submerged in 4% PFA for paraffin embedding and sectioning.

**Measurement of adipocyte size using mean linear intercept**. Adipocyte size in epigonadal WAT was assessed as described previously[46]. Briefly, five-micrometer cross-sections of the PFA-fixed and paraffin-embedded WAT were stained with hematoxylin and eosin as previously described[74]. The mean linear intercept (MLI) of adipocytes was measured in imaged slides (Leica SCN400, Germany; magnification of 20x) by using the program 'cell^D' and a grid pattern of 50 µm to 50 µm (version 5.1, Olympus Europe SE & Co. KG, Hamburg, Germany). Blood vessels and connective tissue were avoided. The MLI was assessed in three to eight random fields of view per WAT section per animal.

**RNA extraction, cDNA synthesis, RNA Sequencing, and quantitative qRT-PCR**. Total RNA was extracted from frozen mouse lung and WAT, murine and human bronchial smooth muscle cells (bSMC), and microdissected lung tissue using TRI Reagent (Sigma-Aldrich, #T9424-200 ml, St. Louis, MO, USA) according to the recommendations. The RNA concentration was measured with Tecan Infinite® M200 PRO NanoQuant and the quality of the RNA was determined by measuring a 260/280 nm-ratio or RNA with the Agilent. Libraries were prepared from 10 ng total RNA using "QuantSeq 3' mRNA-Seq Library Prep Kit FWD for Illumina" (Lexogen) according to the "low input/FFPE/degraded material" protocol. Sequencing was performed in a HiSeq 4000 (SY-401–4001) from Illumina (San Diego, USA). For qRT-PCR experiment, equal amounts of extracted RNA were transcribed into cDNA using Random primer (Roche, #11034731001, Basel,

Switzerland), or Oligo-dT-Primer (Eurofins/MWG, #16-T, Luxemburg), dNTP Mix (Fermentas, #R0193, Waltham, MA, USA), and Murine Leukemia Virus Reverse Transcriptase (MMLV RT) (Promega, #M1705, Fitchburg, WI, USA). In addition, incubation with DNase (Promega, #M6101, Fitchburg, WI, USA) and RNase inhibitor (rRNasin; Promega, # N2515, Fitchburg, WI, USA) was performed to eliminate genomic DNase contamination and block RNase activity, respectively. Subsequently, SYBR Green or TaqMan qRT-PCR was performed using 7500 Real-time PCR system (Applied Biosystem, Foster City, CA, USA). Intron-spanning human and murine primers were designed using NCBI and UCSC Genome Browser database. Primer and TaqMan probes are listed in Supplementary Table 1. Gene expression was calculated with the ΔΔCt-method; β-actin, Gapdh, and 18 s rRNA served as housekeeping genes. List of primers used for qPCR analysis (Supplementary Table 1).

**Protein isolation and immunoblotting.** For protein isolation, mouse tissues and cells were homogenized with lysis buffer [CHAPS buffer (Millipore, #220201, MA, USA) supplemented with a final concentration of 1% Halt protease inhibitor cocktail (Thermo Fisher Scientific, #78442, Waltham, MA, USA)]. Protein concentration was determined using the Pierce™ BCA Protein Assay Kit (Thermo Fisher Scientific, #23225, Waltham, MA, USA) according to the manufacturer's instructions and measured with the Tecan Infinite® M200 PRO NanoQuant. Protein samples were loaded on an 8–12% polyacrylamide gel, separated by electrophoresis (SDS-PAGE), and transferred to a nitrocellulose membrane using semi-dry blotting. After blocking, the membranes were probed with the following antibodies: mouse monoclonal anti-αSMA (Santa Cruz, #sc53142, clone: B4, 1:200, Dallas, TX, USA), monoclonal rabbit anti-pAKT (Cell signaling, #4058, clone 192H12, 1:1000, Danvers, MA, USA), polyclonal rabbit anti-AKT (Cell signaling, #9272; 1:2000, Danvers, MA, USA), polyclonal rabbit anti-pSAP/JNK1 (Thr183/Thr185; Cell signaling, #9251; 1:500, Danvers, MA, USA), polyclonal rabbit anti-SAP/JNK (Cell signaling, #9252; 1:1000, Danvers, MA, USA), monoclonal rabbit anti-p-c-Jun (Ser73; Cell signaling, #3270; clone D47G9, 1:1000, Danvers, MA, USA), monoclonal rabbit anti-c-Jun (Cell signaling, #9165; clone 60A8;1:1000 Danvers, MA, USA), polyclonal rabbit anti-PPARγ (Cell signaling, #2430; 1:1000, Danvers, MA, USA), polyclonal rabbit anti-pSTAT3 (Cell signaling, #9145; 1:1000, Danvers, MA, USA), polyclonal rabbit anti-STAT3 (Cell signaling, #9139; 1:1000, Danvers, MA, USA), polyclonal rabbit anti-p38 (Cell Signalling, #9212, 1:2000, 1:1000, Danvers, MA, USA), monoclonal rabbit anti-pP38 (Cell Signalling, #4511, clone: B3F9, 1:1000, 1:1000, Danvers, MA, USA), whereas monoclonal mouse anti-β-actin (Cell Signaling, #3700, 1:5000, Danvers, MA, USA) served as a loading control. Anti-mouse IgG, HRP-linked (Cell signaling, # 7076, 1:2000, Danvers, MA, USA), and anti-rabbit IgG, HRP-linked (Cell signaling, # 7074, 1:2000, Danvers, MA, USA) were used as secondary antibodies. Densitometric analysis of protein bands was performed using Bio-Rad ImageLab software (Version 5.2.1., Bio-Rad, Munich, Germany). Band intensities from samples were normalized for loading using the β-actin band from the same sample.

**Immunofluorescent staining.** PFA-fixed and paraffin-embedded lung tissue sections (3 μm) were deparaffinized in Neoclear (Sigma-Aldrich, #109843, St. Louis, MO, USA) and rehydrated in a graded ethanol series (100%, 96%, 80% and 70% for 1 min each) to PBS. The tissue was incubated with solution A from the MaxBlock Autofluorescence Reducing kit (MaxVision Biosciences, #MB-L, WA, USA) according to the manufacturer's instructions and washed with 60% EtOH for 1 min and PBS. Antigen retrieval was performed with 10 mM citrate buffer pH6 (Dako, Cat. No. S2369, Santa Clara, CA, USA) at 90–120 °C for 25 min. Sections were blocked with Sea Block for 1 h (Thermo Fisher Scientific, #37527, Waltham, MA, USA), and incubated at 4 °C overnight with polyclonal rabbit anti-FoxO1 (Novusbio, #NB100-2312, 1:400, Centennial, CO, USA), polyclonal rabbit anti-Ki67 (Thermo Fisher Scientific, #PAS19462, 1:1000, Waltham, MA, USA), polyclonal rabbit anti-pSTAT3 (Cell Signaling, #9145; 1:200, Danvers, MA, USA), monoclonal rabbit anti-pAKT (Cell Signaling, #4058, clone 193H12, 1:200, Danvers, MA, USA), p Histone H2A.X (S139, γH2AX; Cell Signaling, #2577, 1:400, Danvers, MA, USA), monoclonal mouse anit-8-Oxo-dG (Abcam, #ab62623, clone 15A3, 1:400, Cambridge, UK) diluted in antibody diluent (DAKO, #S0809, Santa Clara, CA, USA). After primary antibody binding, slides (VWR International, #J1800AMNZ, Radnor, PA, USA) with FoxO1, Ki67, pSTAT3, and pAKT staining were then incubated with monoclonal-Cy3-anti-αSMA (Sigma-Aldrich, #C6198, clone 1A4, 1:200, St. Louis, MO, USA) diluted in antibody diluent (DAKO, #S0809, Santa Clara, CA, USA) at 4 °C overnight. Subsequently, slides were incubated with the respective goat-raised secondary antibody, conjugated with an F488 fluorochrome (Jackson Laboratory, #111-485-003, 1:500, Bar Harbor, ME, USA) for 1 h. Cell nuclei were stained with DAPI (#D9542, Sigma-Aldrich, St. Louis, MO, USA). Tissue sections were incubated with solution B from the MaxBlock Autofluorescence Reducing kit (MaxVision Biosciences, #MB-L, WA, USA) and mounted with glass coverslips using Fluoromount (Sigma-Aldrich, #F4680, St. Louis, MO, USA). Photomicrographs of representative images were taken using a microscope (Olympus BX43, Hamburg, Germany) and a ×40 and ×100 lens using CellSens Dimension software (Version 1.18, Olympus). Quantification of FoxO1, pSTAT3, and pAKT was performed by measuring the respective Integrated Density in αSMA-positive cells. Similarly, the number of proliferating cells was determined by measuring Ki67-positive; αSMA-positive cells. Double-positive cells (Ki67; αSMA) were quantified in 10–20 different fields of view per animal (400x) using Cell D 3.4 Olympus Soft

Imaging Solutions (Olympus). Similarly, quantification of 8-Oxo-dG or γH2AX staining was performed by measuring 8-Oxo-dG Integrated Density or by counting the percentage of γH2AX-positive nuclei relative to all nuclei per field of view.

**Immunohistochemistry.** PFA-fixed and paraffin-embedded lung and WAT tissue sections (3 μm) were deparaffinized in Neoclear (Sigma-Aldrich, #109843, St. Louis, MO, USA) and rehydrated in a graded ethanol series (96%, 70%, and 30% for 5 min each) to H₂O. Antigen retrieval was performed with 1 mM EDTA solution pH 8 at 90–120 °C for 20 min. Afterward, tissue was blocked (Sea Block, Thermo Scientific™, #37527 Waltham, MA, USA) at room temperature for 1 h. Primary antibody polyclonal rabbit anti-CD68 (Abcam, #ab125212, 1:500, Cambridge, UK), monoclonal rabbit anti-CD3 (Thermo Fisher Scientific, #RM-9107-S, Clone SP7, 1:50, Waltham, MA, USA), monoclonal rat anti-CD45 (BD, #550539, Clone 30-F11, 1:25, Franklin Lakes, USA) or monoclonal rat anti-LyG6 (Abcam, #ab2557, 1:50, Cambridge, UK) was applied for 30 min. For LyG6 immunostaining, tissues were pre-treated with Triton 0,3% in PBS for 30 min. Primary antibody incubations were followed by treatments with the secondary antibody anti-rabbit (Histofine® MOUSESTAIN KIT, Nichirei, #414341 F, Japan, rabbit primary antibody) or anti-rat (Histofine® MOUSESTAIN KIT, Nichirei, #414311 F, Japan, rat primary antibody) for 30 min at room temperature. Tissue was then incubated in DAB (Immuno Logic, Bright-DAB-Solution A + B; #BS04-999A + #BS04-999B) for 8 min and subsequently counterstained with hematoxylin and examined by light microscopy using ten random fields of view per lung section. Positive cells were counted and presented as cells per field of view.

**Measurement of the bronchial smooth muscle layer (bSML).** To measure the thickness of the bSML, randomly selected PFA-fixed and paraffin-embedded lung tissue sections (3 μm) were stained for αSMA as a marker of SMC. PFA-fixed and paraffin-embedded lung tissue sections (3 μm) were deparaffinized in Neomount (Kenilworth, #1.090.160.100, NJ, USA) and rehydrated in a graded ethanol series (100%, 96%, 80%, and 70% for one minute each) to PBS. Antigen retrieval was performed using a citrate-based antigen unmasking solution (DAKO, #S2369, 10 mM, pH6, Santa Clara, CA, USA). Sections were blocked with Sea Block (Thermo Fisher Scientific, #37527, Waltham, MA, USA), and incubated with monoclonal mouse anti-αSMA (Santa Cruz, # SC53142, clone: B4 1:200, St. Louis, MO, USA); sections were counterstained with hematoxylin. CellSens Dimensions software (Version 1.18, Olympus) was applied to evaluate bSML thickness. In total, six animals per group and only transversely-sliced bronchi were measured. The thickness of the bSML was analyzed at three different positions and the mean value of the three measurements served as the average bSML thickness.

**Measurement of the vascular remodeling.** To assess vascular remodeling, dual staining for monoclonal mouse anti-αSMA (Santa Cruz, # SC53142, clone: B4 1:200, St. Louis, MO, USA) and polyclonal rabbit anti-von Willebrand factor (vWF; Abcam, #ab6994, 1:200, Cambridge, UK) as an indicator for SMC and endothelial cells, respectively, was performed with paraffin-embedded lung, kidney and liver sections (3 μm). Sections were counterstained with hematoxylin and examined by light microscopy using either ten random fields of view per lung section or a computerized morphometric system (Leica QWin V3, Wetzlar, Germany) for assessing medial wall thickness and the degree (none, partial or complete) of muscularization of small peripheral pulmonary artery (<100 μm). The number of microvessels (<100 μm) was assessed in ten random fields of view per lung section. In total, 3–5 sections per animal were analyzed.

**Cell culture studies.** Primary murine bronchial SMC (mbSMC) were isolated from C57BL/6 (WT), IL-6⁻/⁻, FoxO1^ADA (kindly provided by Dr. T. Wunderlich, Max Planck Institute for Metabolism Research, University Hospital Cologne, Cologne, Germany)[45], SMC^FoxO1 KO (kindly provided by Dr. S. Savai Pullamsetti, Max Planck Institute for Heart and Lung Research, Bad Nauheim, Germany)[34], and respective control mice at P21 or at the age of 10–12 weeks. Human bronchial SMC (hbSMC) (PromoCell; #397Z013.3, #C-12561, Heidelberg, Germany) and human aortal SMC (haSMC) (ATCC, #PCS-100-012, Manassas, VA, USA) were cultured according to the manufacturer's recommendations (PromoCell; Smooth Muscle Cell Growth Medium 2: Lot#427M002; Supplement-Mix: Lot#425M235, Heidelberg, Germany).

**Immunocytochemistry and quantification of pFoxO1, total FoxO1, pFoxO3a, and pSTAT3 in bSMC.** Human and murine bSMCs as well as haSMC were seeded in 24-well plates (7.500 cells per well) on glass coverslips and cultured in a serum-rich medium (10% FBS) for 24 h. Afterward, cells were cultured in serum-reduced medium (2% FBS) overnight, followed by exposure to IL-6 + R with or without Paclitaxel, AKT inhibitor or Stattic for 30 min or 24 h. At the end of the exposure time, cells were fixed in 4% PFA for 15 min and blocked with Sea Block (Thermo Fisher Scientific, #37527, Waltham, MA, USA) for 1 h at room temperature. Cells were then incubated with primary antibodies polyclonal rabbit anti-pFoxO1 (pFoxO1(Thr24)/FoxO3a(Thr32), Cell signaling; #9464; 1:400, Danvers, MA, USA), polyclonal rabbit anti-FoxO1 (Novusbio, #NB100-2312, 1:400, Centennial, CO, USA), polyclonal rabbit anti-pFoxO3a (Ser253, Cell signaling; #9466; 1:400, Danvers, MA, USA) or polyclonal rabbit anti-pSTAT3 (Cell signaling, #9145;

1:200, Danvers, MA, USA) overnight at 4 °C. Subsequently, cells were incubated with the respective goat-raised secondary antibody, conjugated with an F488 fluorochrome (the Jackson Laboratory, #111-485-003, 1:1000, Bar Harbor, ME, USA) for 1 h. Cell nuclei were stained with DAPI (#D9542, Sigma-Aldrich, St. Louis, MO, USA). Quantification of FoxO1, pFoxO1, and pFoxO3a was performed by measuring the Integrated Density in the nuclei and the cytoplasm and were presented as a percentage of total signal per cell. Quantification of pSTAT3 was performed in five images (×10 magnification) per condition by measuring the Integrated Density of total cell and was presented as total signal per cell.

**Assessment of cell proliferation.** Human and murine bSMCs as well as haSMC were seeded into 96-well plates (7.500 cells per well) and cultured in serum-rich medium (10% FBS) for 24 h. Afterward, cells were maintained in a serum-reduced medium overnight (12 h) and then Bromodeoxyuridine (BrdU) labeling solution was added. Next, cells were exposed to IL-1β, leptin, IL-6 + R, Paclitaxel or Stattic. At the end of the 24 h exposure, BrdU incorporation was assayed using a BrdU Cell Proliferation Assay Kit (#11647229001, Roche, Basel, Switzerland) according to the manufacturer's instructions. Alternatively, 4,000 cells were seeded into 24-well plates, cultured in serum-rich medium for a total of 7 days with or without IL-6 + R, insulin or respective vehicle, and counted daily using light microscopy. Proliferation was expressed as the percentage of cells relative to cell count on day 1.

**Assessment of cell viability.** Viability of cultured human and murine bSMC was assessed using MTT assay (ATCC, Manassas, VA; cat. no. 30-1010 K) according to the manufacturer's instruction. Cells were seeded in 96-well plates (7500 cells per well) and grown in serum-rich medium. Afterward, cells were first maintained in serum-reduced medium overnight (12 h) and then exposed to IL-6 + R with or without Paclitaxel and Stattic or vehicle for 24 h. Subsequently, cells were incubated for 2 h with 10 μl of yellow tetrazolium MTT [3-(4, 5-dimethylthiazolyl-2)−2, 5-diphenyltetrazolium bromide], which is reduced by metabolically active cells. Absorbance (570 nM) was measured in a plate reader (Tecan Infinite M200Pro; Männedorf, Switzerland).

**Survival assay.** For Caspase-Glo® 3/7 assay cells were seeded in a 96-well plate (7.500 cells/well) in serum-rich medium. Prior to assessing apoptosis, cells were exposed to IL-6 + R with or without Paclitaxel or vehicle, followed by incubation for one hour with 40 μl of Caspase 3/7 Luciferase Reagent Mix (Caspase-Glo® 3/7 assay; Promega, #G8093, Madison, WI, USA). Total luminescence was measured in a plate reader (Promega GloMax Multi Detection System).

**In vitro Cre-recombinase transduction.** In bronchial SMC from FoxO1^ADA mice, the induction of Cre-mediated rearrangement was accomplished by TAT-Cre-recombinase from Excellgen (Excellgen, #EG-1001, 4 μM, Rockville, MD, USA). Bronchial SMC was incubated with TAT-Cre in serum-free DMEM/F-12 medium (Invitrogen, #11039-047, Carlsbad, CA, USA) with 1% penicillin/streptomycin (Sigma-Aldrich, #P4458-100ML, St. Louis, MO, USA) for 2 h in a 37 °C incubator. The reaction was terminated with the addition of DMEM/F-12 containing 10% FBS (Biochrom, #S0615, Berlin, Germany) and cells were allowed to rest for 10 h.

**Treatment with compounds and drugs.** Human bSMC, murine bSMC or haSMC were maintained in serum-reduced medium (2% FBS) overnight, followed by exposure to IL-1β (Sigma-Aldrich, #SRP8033, 10 ng/ml, St. Louis, MO, USA (vehicle aqua dest)), leptin (R&D System, #498-OB, 100 ng/ml, Minneapolis, MN, USA (vehicle 20 mM Tris-HCL, pH 8.0)), IL-6 (Sigma-Aldrich, #I9646, 100 ng/ml, St. Louis, MO, USA) with soluble IL-6 receptor (R&D Systems, #1830-R, 20 ng/ml, Minneapolis, MN, USA) (vehicle 0.1% BSA)) with or without Paclitaxel [Pax, Sigma-Aldrich, #T7402, 1 μM, St. Louis, MO, USA (vehicle ethanol)], Huminsulin (Eli Lilly Canada Inc., #HI0319, 10 μg/ml, Toronto, Ontario, Canada (vehicle 0,9% NaCl)), AKT Inhibitor (MK-2206, Selleck, # S1078, 5 μM, Houston, TX, USA) (vehicle DMSO)), Stattic (Millipore, #573099, 10 μM, Burlington, MA, USA (vehicle ethanol)), and FoxO1 inhibitor (AS1842856; Merck, Calbiochem, # 344355, 1 μM, Darmstadt, Germany (vehicle DMSO)) for 30 min or 24 h.

**Human study population.** The UK Biobank is a longitudinal cohort study of >500,000 individuals aged 40–69 years initiated in the UK in 2006–2010. The UK Biobank organization collected data from a baseline study visit as well as administrative data from inpatient and outpatient health systems on all participants. Other than the data collected and provided by the UK Biobank organization, no additional data on UK Biobank participants was obtained by the researchers for the purposes of this study. For our analyses, we excluded 1756 individuals due to Chronic Obstructive Pulmonary Disease (COPD), and 52,990 current smokers. After these exclusions, 448,351 individuals were eligible for observational analyses. The exposures of interest for our main analysis were forced expiratory volume in 1 s/forced vital capacity (FEV1/FVC) ratio Z-score (field ID 20258), and comparative body size at 10 years of age (field ID 1687, recoded to two groups: about average and thinner=0; plumper =1). Asthma was defined as International Classification of Diseases (ICD) edition 10 codes J45. Details of these measurements can be found in the UK Biobank Data Showcase (http://biobank.ctsu.ox.ac.uk/crystal/).

**Ethical statement.** This study includes information already collected within the UK Biobank. No new biological materials or other data from the participants were collected. UK Biobank has consulted widely not only with the scientific community, but also with the public, its participants, and other interested parties. The UK Biobank is overseen by an independent committee, the UK Biobank Ethics Advisory Committee (EAC). The EAC is a Committee of the UK Biobank Board who's remit is to provide advice to the Board on ethical issues that arise during the maintenance, development, and use of UK Biobank, including identifying, defining, and examining relevant ethical issues. The UK Biobank organization obtained informed consent for the participation of all participants and collection of data for distribution and use in biomedical research included in this manuscript. The informed consent procedures can be found here https://www.ukbiobank.ac.uk/media/0xsbmfmw/egf.pdf. The institutional review board at Stanford University was consulted and acknowledged that the double-blinded nature of data collection and third-party distribution of the UK Biobank organization creates a dataset of no risk or minimal risk to research participants. For this reason, all work with these data qualifies as exempt from IRB review.

**Observational analyses in the UK biobank.** We performed multivariable logistic and linear regression models to determine associations of asthma and FEV1/FVC ratio Z-score with comparative body size at 10 years of age. All association analyses were adjusted for age, sex, region of the UK Biobank assessment center (England, Scotland, and Wales), ethnicity (white, black, Asian, mixed), and body mass index. Regressions analyses were conducted with the R software.

**Analysis of the data.** Values for murine studies and cell culture studies are shown as means ± standard error of the mean (SEM). Two-way ANOVA or one-way ANOVA followed by a Bonferroni post-test, two-sided Mann–Whitney test, and two-sided Student's $t$ test were used to test for statistical significance at the given time points. For transcriptomic analysis, $P$ values are calculated using Wald test and $P$-adjusted values using the FDR/Benjamini–Hochberg approach. A $P$ value <0.05 was considered statistically significant.

**Reporting summary.** Further information on research design is available in the Nature Research Reporting Summary linked to this article.

## Data availability

The RNA sequencing data have been deposited in GSA with the accession number: CRA007218[75]. All other data supporting the findings of this study are available within the article and its Supplementary Files. This research has been conducted using the UK Biobank Resource under Application Number 13721. We used human data collected at the UK Biobank assessment centers (http://biobank.ctsu.ox.ac.uk/crystal/) field ID 20258 and field ID 1687. Source data are provided with this paper.

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

## Acknowledgements

The authors acknowledge the excellent technical assistance of Uta Eule and Ewa Bieniek (Max Planck-Institute for Heart and Lung Research, Bad Nauheim, Germany), Malte Heykants [Comparative Medicine, Center for Molecular Medicine, University of Cologne (CMMC), faculty of medicine and university hospital, Cologne, Germany] as well as Dr. Christian Jüngst [imaging facility, Cologne Excellence Cluster on Cellular Stress Responses in Aging-Associated Diseases (CECAD), Center for Molecular Medicine Cologne, University of Cologne, Cologne, Germany]. We thank Prof. Dr. Baldus and Dr. Martin Mollenhauer [Clinic III for Internal Medicine, Department of Cardiology, Center for Molecular Medicine Cologne (CMMC), University of Cologne, University Hospital Cologne, Cologne, Germany] for kindly providing us human aortic SMCs. This research has been conducted using the UK Biobank Resource under Application Number 13721; we greatly thank Prof. Dr. Themistocles Assimes (Department of Medicine, Cardiovascular Medicine, Stanford University, Stanford, USA) for providing us access to the UK Biobank Resource. This work was supported by Deutsche Forschungsgemeinschaft [AL1636/2-1 (M.A.A.A.); AL1636/5-1; SFB 1213 (Project A01, A05; S.S.P.)]; Marga und Walter Boll Stiftung (210-02-16; M.A.A.A.); Stiftung Oskar-Helene-Heim (V.J.; M.A.A.A.); Center for Molecular Medicine Cologne, faculty of medicine and university hospital Cologne, Germany (CMMC; M.A.A.A.); Köln Fortune, faculty of medicine and university hospital Cologne, Germany (283/2017; KD; M.A.A.A.); Graduate Program "Pharmakologie und Therapieforschung", center of pharmacology, university hospital Cologne, Germany (JW); intramural funding, Comparative Medicine, Center for Molecular Medicine, University of Cologne (CMMC), faculty of medicine and university hospital Cologne, Germany (E.M.); BHF Intermediate Basic Science Research fellowship (FS/15/59/31839; C.J.R.) and Academy of Medical Sciences Springboard fellowship (SBF004\1095; C.J.R.).

## Author contributions

J.S., K.D., V.J., D.Z., J.P., S.S.P., and M.A.A.A. conceived and designed the research; J.S., K.D., V.J., D.Z., J.W., T.G., C.V., R.W., B.K., J.M., T.N., E.M., S.D., and M.A.A.A. performed experiments; J.S., K.D., V.J., D.Z., J.W., T.G., B.K., J.M., E.M., S.S.P., and M.A.A.A. analyzed data; J.S., K.D., V.J., D.Z., T.G., B.K., O.K., J.M., S.v.K.R., C.J.R., A.U., D.H., M.O., E.M., S.N., S.D., T.W., J.P., W.S., J.D., S.S.P., and M.A.A.A. interpreted results of experiments; J.S., K.D., V.J., S.S.P., and M.A.A.A. drafted the manuscript; J.S., K.D., V.J., E.M., W.S., J.D., S.S.P., and M.A.A.A. edited and revised manuscript. J.S., K.D., V.J., D.Z., J.W., T.G., C.V., R.W., B.K., O.K., J.M., S.v.K.R., C.J.R., A.U., D.H., T.N., M.O., E.M., S.N., S.D., T.W., J.P., W.S., J.D., S.S.P., and M.A.A.A. approved the final version of the manuscript.

## Funding

## Competing interests

The authors declare no competing interests.

## Additional information

¹Faculty of Medicine and University Hospital Cologne, Translational Experimental Pediatrics–Experimental Pulmonology, Department of Pediatric and Adolescent Medicine, University of Cologne, Cologne, Germany. ²Faculty of Medicine and University Hospital Cologne, Center for Molecular Medicine Cologne (CMMC), University of Cologne, Cologne, Germany. ³Division of Cardiovascular Medicine, Department of Medicine, Stanford University School of Medicine, Stanford, CA, USA. ⁴Stanford Cardiovascular Institute, Stanford University, Stanford, CA, USA. ⁵Faculty of Medicine and University Hospital Cologne, Cologne Center for Genomics (CCG), University of Cologne, Cologne, Germany. ⁶Faculty of Medicine and University Hospital Cologne, Department of Pediatric and Adolescent Medicine, University of Cologne, Cologne, Germany. ⁷Institute for Lung Health (ILH), University of Giessen and Marburg Lung Centre (UGMLC), Member of the German Centre for Lung Research (DZL), Gießen, Germany. ⁸Faculty of Medicine and University Hospital Cologne, Pediatric Pulmonology, Department of Pediatric and Adolescent Medicine, University of Cologne, Cologne, Germany. ⁹National Heart and Lung Institute, Hammersmith Campus, Imperial College London, London, UK. ¹⁰Faculty of Medicine and University Hospital Cologne, Institute of Pathology, University of Cologne, Cologne, Germany. ¹¹Faculty of Medicine and University Hospital Cologne, Comparative Medicine, Center for Molecular Medicine Cologne (CMMC), University of Cologne, Cologne, Germany. ¹²Department of Lung Development and Remodeling, Max-Planck-Institute for Heart and Lung Research, Member of the German Center for Lung Research (DZL), Bad Nauheim, Germany. ¹³Max-Planck-Institute for Metabolism Research, Cologne, Germany. ¹⁴Faculty of Medicine and University Hospital Cologne, Cologne Excellence Cluster for Stress Responses in Ageing-Associated Diseases (CECAD), University of Cologne, Cologne, Germany. ¹⁵Department of Internal Medicine, German Center for Lung Research (DZL), Cardio-Pulmonary Institute (CPI), Justus Liebig University, Giessen, Germany. ¹⁶These authors contributed equally: Jaco Selle, Katharina Dinger, Vanessa Jentgen. ✉email: miguel.alejandre-alcazar@uk-koeln.de

