## [Peer Review File · Nature Communications]

Maternal and perinatal obesity induce bronchial obstruction and pulmonary hypertension via IL-6-FoxO1-axis in later lifeREVIEWER COMMENTS

Reviewer #1 (Remarks to the Author):

[redacted]

Reviewer #2 (Remarks to the Author):

The authors study metabolic programming and provide compelling evidence to demonstrate that a maternal high fat diet results in overweight offspring with evidence of bronchial and pulmonary arterial smooth muscle thickening. This is attributed to increased IL6 levels that regulate Foxo1 and Stat3 signaling. Clinical relevance is established through GWAS and mendelian randomization analyses that identify a relationship between FoxO1 and childhood obesity and decreased lung function. This is an understudied area, the experimental plan is rational, and the findings intriguing. Some comments for consideration:

1. It's understood that the focus of the study is remodeling of the bronchi and pulmonary arteries, but the question arises as to whether or not metabolic patterning also affects the systemic vasculature and, if so, hypertension contributes to the findings.
2. Are there developmental abnormalities in the bronchi and the bronchial-pulmonary artery relationship in offspring of high-fat dams? Is the increase in airway resistance attributable to bronchial constriction (from thickened SMC) or structural abnormalities or both?
3. Part of the experimental design is based on the premise that IL6 is the main proinflammatory mediator eliciting vascular remodeling and this is responsible for FoxO1 mediated effects. What about other proinflammatory adipocytokines that are increased by the high fat diet and do any of these affect SMC proliferation?
4. In the RNASeq analysis, was FoxO1, Stat3, or Jun expression up- or down-regulated?
5. The presumption is that Stat3 activates FoxO1, but there are many examples where FoxO1 blocks Stat3. Furthermore, there is plenty of evidence that each of the TFs acts independent of the other. What is the evidence that this is a unidirectional pathway as you illustrate in Fig. 8 and through the experiments?
6. What about other molecular activators of FoxO1? No change in the RNASeq analysis doesn't negate the involvement of other transcription factors, especially since the analysis includes putative and confirmed sites. If restricted to only confirmed TF sites, what other TFs are important, how does this change the ratio of genes and what regulates them?
7. Do you detect changes in the most highly FoxO1 or Stat3 regulated proteins in the smooth muscle cells?
8. Analyses implicate FoxO1, Stat3, and Jun in smooth muscle cell remodeling, but only FoxO1 is evaluated in many of the subsequent models. The reason for this isn't clear and Stat3 and Jun signaling should be examined. This is especially important as it's not clear what the contribution is of proteins regulated by each of the transcription factors to the SMC phenotype.
9. As an example of #7, you report that IL6 has no effect on SMC-specific FoxO1 KO proliferation, but what about on Stat3?
10. Are the differences in inflammatory cell infiltration in the models given FoxO1 expression in T and B cells?
11. It's difficult to appreciate the co-IF staining of vessels in many of the images. The addition of higher magnification inserts would be helpful for the reader.

Reviewer #3 (Remarks to the Author):

General Comment

This is a very elegant and highly innovative study on metabolic programming in the context of chronic lung disease. Specifically, the role of maternal and perinatal obesity for vascular remodelling and smooth muscle cell proliferation predisposing to chronic lung disease beyond infancy has been explored. By establishing a murine model of metabolic programming the authors aimed to show if the adipose tissue-IL-6-FOXO1 axis might be responsible for bronchial obstruction and pulmonary hypertension later in life. Overall, this is an excellent study providing compelling mechanistic evidence for FOXO1 being a key player in metabolic programming of chronic lung disease. Whereas this is a very strong part of the paper, the exclusive allocation of FOXO1 dysregulation to IL-6 signaling is less convincing. Additional players like leptin and insulin need to be considered and discussed.

Specific comments

1. As shown in Fig. 1 E and F, leptin is also substantially increased and this adipokine is also known to sequester FOXO1 to the cytoplasm (via Akt, but JAK/STAT signaling may also be involved). This is not further pursued or discussed and needs to be addressed by the authors.
2. Fig. 1 D: The authors claim insulin resistance in their mouse model, data on circulating insulin should be presented. High insulin levels would be an additional player to affect FOXO1 localisation due to Akt-mediated phosphorylation and cytosolic retention.
3. At P21, additional circulating cytokines might be monitored that could also interfere with FOXO1 function.
4. Fig. 6 A-L: The authors use IL-6 global KO mice to provide evidence that IL-6 is mediating the bronchial and vascular wall remodeling after perinatal high-fat diet. Although the in vitro experiments clearly show that IL-6 is able to induce the FOXO1-mediated proliferation, the KO mice data are not completely convincing for the following reasons: Circulating leptin is not monitored and data on insulin are lacking. Further, the adipose tissue from the KO mice needs to be better characterised. What is the cell size, is there inflammation, what about adiponectin? Within adipose tissue, IL-6 plays an important role for crosstalk with macrophages. Hence, in the KO setting, the physiology of adipose tissue may be altered and this needs to be taken into account.

Reviewer #4 (Remarks to the Author):

I was asked to specifically review the aspect of Mendelian randomization of the manuscript, hence my comments will only pertain this single aspect. The 2-sample MR analysis is correctly performed and the results are convincing. There is one weakness of the approach: the full sample overlap. This biases the results towards the observational correlation and since the instruments were selected from a single sample, they are subject to winner's curse and the bias is not straightforward to be estimated (environmental covariance / $E(F\text{-statistic})$), since the $E(F\text{-stat})$ will be heavily overestimated, hence the bias will be underestimated. Since the authors have access to the UKB data (as they used it for the observational analysis), they could compute the summary stats for a random subsample for body size at age 10 and for the remaining sample for lung function. Alternatively, they could use the sex-specific results from the Neale lab: use women only for the exposure GWAS and men-only for the outcome GWAS (or the other way around). This could confirm that there is no serious bias in the MR estimates due to sample overlap.

Also, couldn't the authors use birth weight GWAS summary stats as another proxy for perinatal obesity?

We greatly appreciate the reviewers' comments, which we included in the revised manuscript. Please find our point-by-point reply to each comment below.

Reviewer 1:

[redacted]

Reviewer 2:

1. It's understood that the focus of the study is remodeling of the bronchi and pulmonary arteries, but the question arises as to whether or not metabolic patterning also affects the systemic vasculature and, if so, hypertension contributes to the findings.

We appreciate this important question regarding systemic effects of maternal and perinatal obesity on systemic vasculature. Prior studies from our group showed that maternal and perinatal obesity adversely affect renal sodium excretion and increases renal matrix deposition; however, glomerular function was not affected (5). The present study shows bronchial and lung vascular remodeling after maternal and perinatal obesity. To further address systemic consequences, we measured left ventricular pressure as an indicator of systemic hypertension and found no differences between HFD and SD at P70 [Fig. 2L, page 6 (lines 167-168)].

2. Are there developmental abnormalities in the bronchi and the bronchial-pulmonary artery relationship in offspring of high-fat dams? Is the increase in airway resistance attributable to bronchial constriction (from thickened SMC) or structural abnormalities or both?

The reviewer points out an important aspect, namely aberrant bronchial and vascular development following maternal and perinatal obesity. To this end, we performed von Willebrand staining (vWF) and quantified microvessels with a diameter of 0 - 20 μm and 20 - 100 μm . While muscularization and medial wall thickness was increased (Fig. 2E, F), the number of microvessels was not different between HFD and SD (Suppl. Fig. 2C). To further study if maternal and perinatal obesity do not only induce bronchial muscularization and increased airway resistance we performed lung function test and exposed mice to methacholine in increasing doses. The lung function analysis did not reveal a hyperreagibility [Suppl. Fig. 2A, B, page 6 (lines 154-156 and lines 160-161)].

3. Part of the experimental design is based on the premise that IL6 is the main proinflammatory mediator eliciting vascular remodeling and this is responsible for FoxO1 mediated effects. What about other proinflammatory adipocytokines that are increased by the high fat diet and do any of these affect SMC proliferation?

White adipose tissue (WAT) is a metabolically highly active endocrine organ that secretes a variety of adipocytokines. In the present study, gene expression and serum concentrations of leptin and IL-6 were elevated at P21, but not at P70 in WT^{HFD} when compared to WT^{SD} (Fig. 1E, F). Other adipocytokines include adiponectin, leptin, IL-1 β , TNF α , Visfatin and Resistin. Initial studies assessed gene expression of these adipocytokines at P21 and did not find any significant differences in gene expression using qRT-PCR between WT^{HFD} and WT^{SD}. In the revised manuscript, we included gene expression or serum concentrations of adiponectin, leptin, and IL-1 β as representative adipocytokines.

We did not detect an increase in adiponectin or IL-1 β after WT^{HFD} when compared to WT^{SD} at P21 or P70. Moreover, we measured these adipocytokines in IL-6^{-/-} mice at P21 and found a similar expression pattern as in WT after HFD. Interestingly, gene expression of adiponectin and IL-1 β in WAT was lower in IL-6^{-/-} than in WT. Moreover, leptin expression was elevated, whereas adiponectin and IL-1 β were not regulated in IL-6^{-/-} HFD when compared to IL-6^{-/-} SD. These additional data are shown in Suppl. Fig. 1. [page 5 (lines 136-139) and page 12 (lines 307-308)]. We also compared adipocyte size of WT^{SD} and IL-6^{-/-} SD and found a mild adipocyte hypertrophy in IL-6^{-/-} SD. While prior studies from our group showed a mild adipocyte hypertrophy in epigonadal WAT of WT^{HFD} mice compared to WT^{SD} (6), IL-6^{-/-} were protected from this effect. We referred to this data and our previous study on pages 11 and 12 (lines 303-308) (Suppl. Fig. 8A-D).

To determine if other adipocytokines regulate SMC proliferation we exposed cultured primary murine bronchial SMCs to leptin or IL-1 β , we did not find a significant effect on proliferation [Fig. 7A, B; page 10 (lines 255-256)].

4. In the RNASeq analysis, was FoxO1, Stat3, or Jun expression up- or down-regulated?

Foxo1, Stat3 and Jun expression were not altered in RNASeq, possibly due to low n-number (3 per group). Therefore, we isolated primary murine bronchial smooth muscle cells (bSMC) from HFD and SD, followed by measurement of these transcription factors. Gene expression of Stat3 was increased, FoxO1 mRNA was decreased while Jun expression was unaltered in bSMC^{HFD} when compared to bSMC^{SD} [Fig. 3 E, page 7 (lines 195-198); Fig. 4O, page 9 (lines 241-243)]. In addition, we performed immunoblots from total lung homogenates of HFD and SD at P21 and found an increased and unaltered protein abundance of Stat3 and c-Jun, respectively, when HFD was compared to SD [Fig. 3F; page 7 (lines 195-198)].

5. The presumption is that Stat3 activates FoxO1, but there are many examples where FoxO1 blocks Stat3. Furthermore, there is plenty of evidence that each of the TFs acts independent of the other. What is the evidence that this is a unidirectional pathway as you illustrate in Fig. 8 and through the experiments?

A possible feedback mechanism from FoxO1 to Stat3 is an interesting and important aspect raised by the reviewer. To address this research question, we measured phosphorylated Stat3 in four approaches: first, in total lung homogenates of genetically modified mice with a FoxO1 ablation in smooth muscle cells (SMC^{FoxO1^{-/-} Ko}) [Fig. 8F; page 12 (lines 332-333)]; second, in cultured bSMCs from SMC^{FoxO1^{-/-} Ko} [Fig. 5P, page 10 (lines 275-277)]; third, in bSMC exposed to a FoxO1 inhibitor [Fig. 5M: page 10 (lines 271-272)]; and fourth, in lungs of WT mice treated with Paclitaxel, a FoxO inhibitor [Suppl. Fig. 9A; page 13 (lines 357-360)]. We did not detect an inhibition or activation of Stat3 *in vivo* or *in vitro* after inhibition or activation of FoxO1. These results propose an unidirectional axis of Stat3- FoxO1 in the setting of maternal and perinatal obesity induced bronchial obstruction and pulmonary hypertension.

6. What about other molecular activators of FoxO1? No change in the RNASeq analysis doesn't negate the involvement of other transcription factors, especially since the analysis includes putative and confirmed sites. If restricted to only confirmed TF sites, what other TFs are important, how does this change the ratio of genes and what regulates them?

We assessed additional transcription factors and regulators of FoxO1, including c-Jun, PPAR γ , JNK, and P38. While mRNA, protein and phosphorylation of Stat3 was significantly upregulated in total

lung homogenate after HFD or in bronchial and vascular SMC at P21 [Fig. 3E, F, **page 7 (lines 195-198)**; Fig. 4A-D, **page 8 (lines 216-220)**], other molecules (c-Jun, PPAR γ , JNK, and P38) were not regulated after maternal and perinatal obesity or after FoxO1 activation (Paclitaxel) (Fig. 3E, F, **page 7 (lines 195-198)**; and Suppl. Fig. 5, 9, **page 8 (lines 220-222)** and **page 13 (lines 357-360)**). Further analysis of other transcription will be required to understand the complex network of transcription factors in the lung.

7. Do you detect changes in the most highly FoxO1 or Stat3 regulated proteins in the smooth muscle cells?

We measured *Socs3* expression in bronchial smooth muscle cells (bSMC) isolated from offspring of high fat diet (HFD-) or standard diet (SD-) fed dams and found a significant increase along with higher *Stat3* mRNA in bSMC^{HFD} when compared to bSMC^{SD} [Fig. 3, **page 7 (lines 195-198)**]. *Socs3* is a downstream target of Stat3 signaling and elevated *Socs3* gene expression indicates Stat3 activation (7). Assessment of pStat3 confirmed an activation of Stat3 signaling in HFD when compared to SD [Fig. 4A-D, **page 8 (lines 216-220)**]. Moreover, we assessed in *Foxo3*, *Foxo4*, and *FoxO6* as well as FoxO1 target genes (*Cdkn1b*, *Bcl6*, and *Gadd45a*) in bSMC^{HFD} and bSMC^{SD} and found no slight regulation of *Foxo4* along with a down regulation of *Cdkn1b*, *Bcl6*, and *Gadd45a* [Fig. 4O, P, **page 9 (lines 241-243)**]. Unfortunately, based on a limited amount of mice and bSMC we could not assess protein abundance,

8. Analyses implicate FoxO1, Stat3, and Jun in smooth muscle cell remodeling, but only FoxO1 is evaluated in many of the subsequent models. The reason for this isn't clear and Stat3 and Jun signaling should be examined. This is especially important as it's not clear what the contribution is of proteins regulated by each of the transcription factors to the SMC phenotype.

This is an important aspect raised by the reviewer. We have measured additional transcription factors, including c-Jun, PPAR γ , JNK, and P38 in lungs of mice at P21 and P70 after maternal and perinatal obesity or treatment with Paclitaxel [Fig. 3E, F, **page 7 (lines 194-198)**; and Suppl. Fig. 5, 9, **page 7 (lines 220-222)** and **page 13 (lines 357-360)**]. The analyses did not show an effect of HFD on activation of these pathways or on the transcription factors. In contrast, *Stat3* mRNA was significantly increased in bSMC after maternal and perinatal obesity (HFD) when compared to SD. Complementary immunoblot and co-immunofluorescent staining showed a significant activation of Stat3 signaling in total lung homogenates and in bronchial and vascular SMCs at P21, respectively [Fig. 4A-D, **page 8 (lines 216-220)**]. In addition, analyses of FoxO1 *in vivo* and *in vitro* confirmed an IL-6-Stat3-FoxO1 signaling through which IL-6 inactivates FoxO1 in SMCs and promotes proliferation (Fig. 4, 5, 6). Therefore, we focussed in the present study on Stat3 and FoxO1, but future studies should elucidate alternative transcription factors.

9. As an example of #7, you report that IL6 has no effect on SMC-specific FoxO1 KO proliferation, but what about on Stat3?

To further strengthen our notion that IL-6-Stat3-mediated proliferation of bSMC requires FoxO1 we exposed bSMC from genetically modified mice with a SMC-specific ablation of FoxO1 (SMC^{FoxO1 Ko}) to IL-6/sIL-6R and assessed Stat3 activation, proliferation, and viability. SMC^{FoxO1 Ko} exhibited increased proliferation and viability, but no changes in phosphorylated Stat3 when compared bSMC from control mice. Treatment with IL-6/sIL-6R activated Stat3 significantly, but did not increase

proliferation and viability, supporting the notion that IL-6-Stat3 requires FoxO1 to promote proliferation of bSMC [Fig. 5N, O, P, **page 10 (lines 272-279)**].

[redacted]

10. Are the differences in inflammatory cell infiltration in the models given FoxO1 expression in T and B cells?

We addressed this important research by immunohistochemical staining lungs after HFD and SD at P21 and P70 using CD3, CD45, and LyG6 as markers for T cells, leucocytes (B cells) as well as granulocytes. The analyses did not show any immune cell infiltration after maternal and perinatal obesity [Suppl. Fig. 3, **pages 7 (lines 174-178)**].

11. It's difficult to appreciate the co-IF staining of vessels in many of the images. The addition of higher magnification inserts would be helpful for the reader.

We agree with the reviewer and have exchanged the images with a higher quality ones: **Fig. 4E, F, G, H; Fig. 7 K,L, M, N; Fig. 9E, J, K.**

**Reviewer 3:
Specific comments**

1. As shown in Fig. 1 E and F, leptin is also substantially increased and this adipokine is also known to sequester FOXO1 to the cytoplasm (via Akt, but JAK/STAT signaling may also be involved). This is not further pursued or discussed and needs to be addressed by the authors.

We appreciate the comment. Indeed, leptin and IL-6 share similar signaling pathways, including Stat3. Interestingly, IL-6^{-/-} mice exhibited increased leptin mRNA expression, but were protected from FoxO1 inactivation and subsequent bronchial and vascular remodeling [Fig. 1E, F, G, H, **page 5 (lines 125-136)**; Suppl. Fig. 1A, C, **page 5 (lines 136-139)**]. These findings indicate that the regulation of FoxO1 in bSMC after maternal and perinatal obesity is likely mediated through IL-6-Stat3 and less through leptin-Stat3. As employing Leptin deficient mice, and exploring the regulation of STAT3 needs extensive investigations, we addressed this important comment in the discussion.

Moreover, we assessed Akt signaling in total lung homogenates at P21 and P70 and did not find any differences between HFD and SD (Suppl. Fig. 5, 9). These data are outlined on **page 8 (lines 220-222)** and on **page 13 (lines 357-360)**.

2. Fig. 1 D: The authors claim insulin resistance in their mouse model, data on circulating insulin should be presented. High insulin levels would be an additional player to affect FOXO1 localisation due to Akt-mediated phosphorylation and cytosolic retention.

Insulin is an important regulator of cell homeostasis and has been shown in prior studies from our group to be increased in mice at P21 after maternal and perinatal obesity, supporting the notion of insulin resistance (5). Interestingly, assessment of pAkt signaling as the main downstream of insulin at P21 did not show an activation or inhibition of Akt in lungs after HFD [Suppl. Fig. 5, **page 8 (lines 220-222)**; and Suppl. Fig. 9, **page 13 (lines 357-360)**]; in contrast IL-6-Stat3 signaling was markedly activated in lungs at P21 after maternal and perinatal obesity [Fig. 4A-D, **page 8 (lines 216-220)**]. These findings allowed us to pursue the IL-6-Stat3-FoxO1 signaling in subsequent experiments.

3. At P21, additional circulating cytokines might be monitored that could also interfere with FOXO1 function.

Since our focus was primarily cytokines related to white adipose tissue (WAT), we focussed on adipocytokines. The WAT is a highly active endocrine organ secreting a variety of cytokines, including IL-6 or leptin. We measured additional adipocytokines and included adiponectin and IL-1 β as key representatives in the present study. Gene expression and serum concentrations of both adipocytokines were unaltered or rather reduced in mice after maternal and perinatal obesity [Suppl. Fig. 1, **page 5 (lines 136-139)**].

Moreover, in contrast to IL-6, exposure of murine bronchial smooth muscle cells (bSMC) to IL-1 β or leptin did not significantly regulate FoxO1 or proliferation [Suppl. Fig. 7, **page 10 (lines 255-256)**]. Collectively, *in vivo* and *in vitro* data highlight the important role of elevated IL-6 in the regulation of FoxO1 and cell proliferation of bSMC.

4. Fig. 6 A-L: The authors use IL-6 global KO mice to provide evidence that IL-6 is mediating the bronchial and vascular wall remodeling after perinatal high-fat diet. Although the *in vitro* experiments clearly show that IL-6 is able to induce the FOXO1-mediated proliferation, the KO mice data are not completely convincing for the following reasons: Circulating leptin is not monitored and data on insulin are lacking. Further, the adipose tissue from the KO mice

needs to be better characterised. What is the cell size, is there inflammation, what about adiponectin? Within adipose tissue, IL-6 plays an important role for crosstalk with macrophages. Hence, in the KO setting, the physiology of adipose tissue may be altered and this needs to be taken into account.

We appreciate the important comment by the reviewer. To address these important aspects additional experiments were performed: first, we studied white adipose tissue (WAT) from IL-6^{-/-} and WT mice. Quantitative histomorphometric analyses of WAT showed that the mean linear intercept (MLI) as an indicator of adipocyte size was higher in IL-6^{-/-} SD than in WT^{SD}, indicating an adipocyte hypertrophy due to IL-6 deficiency; whereas exposure of IL-6^{-/-} to HFD did not affect adipocyte size when compared to SD (Suppl. Fig. 8A-C). We also stained for CD68 as a marker of macrophages, but did not find differences in IL-6^{-/-} after maternal and perinatal obesity (HFD) when compared to SD (Suppl. Fig. 8D). These data are described on **pages 11-12 (lines 303-308)**. We next measured the gene expression of adipocytokines such as adiponectin and IL-1 β in WAT and found a lower expression in IL-6^{-/-} when compared to WT at P21 and P70; however, the response of WAT to HFD with regard to the expression of adipocytokines was similar in IL-6^{-/-} and WT mice, with no changes in adiponectin and IL-1 β mRNA, but a significant increase in leptin mRNA and serum leptin levels [Suppl. Fig. 1A, C, **page 12 (lines 307-308)**; and Fig. 7B, **page 11 (lines 299-302)**]. Despite a similar response to HFD and increased leptin, IL-6^{-/-} mice were protected from inactivation of FoxO1 and vascular as well as bronchial remodeling after maternal and perinatal obesity. The key role of IL-6 amongst other adipocytokines in this process was further strengthened by the fact that exposure of bSMC to IL-1 β or leptin does not affect FoxO1 or proliferation [Suppl. Fig. 7, **page 10 (lines 255-256)**].

Reviewer 4:

I was asked to specifically review the aspect of Mendelian randomization of the manuscript, hence my comments will only pertain this single aspect. The 2-sample MR analysis is correctly performed and the results are convincing. There is one weakness of the approach: the full sample overlap. This biases the results towards the observational correlation and since the instruments were selected from a single sample, they are subject to winner's curse and the bias is not straightforward to be estimated (environmental covariance / E(F-statistic)), since the E(F- stat) will be heavily overestimated, hence the bias will be underestimated. Since the authors have access to the UKB data (as they used it for the observational analysis), they could compute the summary stats for a random subsample for body size at age 10 and for the remaining sample for lung function. Alternatively, they could use the sex-specific results from the Neale lab: use women only for the exposure GWAS and men-only for the outcome GWAS (or the other way around). This could confirm that there is no serious bias in the MR estimates due to sample overlap.

Also, couldn't the authors use birth weight GWAS summary stats as another proxy for perinatal obesity?

We appreciate the important comments by the Reviewer. We agree with the crucial point highlighted by the Reviewer and we assessed it as a limitation in the manuscript at page 19: We are limited to using measures currently available in the UK Biobank since we did not have access to external summary statistics. Consequently, our results can be biased because of the participants overlapping.

A recent study from our group used birth weight GWAS summary statistics as exposure and lung function (FVC and FEV1) as outcome. This study was recently accepted by Scientific Reports. In this

previous study, we provided evidence for a causal effect between low birth weight in humans and reduced lung function in adulthood. We used the GWAS summary statistics of birth weight performed by the Early Growth Genetics Consortium as exposures and the GWAS summary statistics of FEV1 and FVC from the UK Biobank as outcome. In order to support our finding we cited our recently accepted previous article along with Warrington et al., *Nature Genetic*, 2019 (8) in the present manuscript.

Page 19, lines 517-529: *We are limited to using measures currently available in the UK Biobank since we did not have access to external summary statistics. Consequently, our results can be biased because of the participants overlapping. To validate our findings, another recent paper published by our group (manuscript accepted, Scientific Reports) assessed a causal effect between low birth weight (proxy for perinatal obesity) and reduced lung function in adulthood. In this previous publication we used the GWAS summary statistics of birth weight performed by the Early Growth Genetics Consortium as exposures and the GWAS summary statistics of FEV1 and FVC from the UK Biobank as outcome (Warrington et al, Nature Genetic, 2019).*

Reference:

1. K. Leppek, R. Das, M. Barna, Functional 5' UTR mRNA structures in eukaryotic translation regulation and how to find them. *Nat Rev Mol Cell Biol* **19**, 158-174 (2018).
2. A. Johansson, M. Rask-Andersen, T. Karlsson, W. E. Ek, Genome-wide association analysis of 350 000 Caucasians from the UK Biobank identifies novel loci for asthma, hay fever and eczema. *Hum Mol Genet* **28**, 4022-4041 (2019).
3. W. J. Astle *et al.*, The Allelic Landscape of Human Blood Cell Trait Variation and Links to Common Complex Disease. *Cell* **167**, 1415-1429 e1419 (2016).
4. M. H. Chen *et al.*, Trans-ethnic and Ancestry-Specific Blood-Cell Genetics in 746,667 Individuals from 5 Global Populations. *Cell* **182**, 1198-1213 e1114 (2020).
5. P. Kasper *et al.*, Renal Metabolic Programming Is Linked to the Dynamic Regulation of a Leptin-Klf15 Axis and Akt/AMPKalpha Signaling in Male Offspring of Obese Dams. *Endocrinology* **158**, 3399-3415 (2017).
6. T. Litzenburger *et al.*, Maternal high-fat diet induces long-term obesity with sex-dependent metabolic programming of adipocyte differentiation, hypertrophy and dysfunction in the offspring. *Clin Sci (Lond)*, (2020).
7. A. Isobe *et al.*, STAT3-mediated constitutive expression of SOCS3 in an undifferentiated rat trophoblast-like cell line. *Placenta* **27**, 912-918 (2006).
8. N. M. Warrington *et al.*, Maternal and fetal genetic effects on birth weight and their relevance to cardio-metabolic risk factors. *Nat Genet* **51**, 804-814 (2019).

REVIEWER COMMENTS

Reviewer #1 (Remarks to the Author):

[redacted]

Reviewer #2 (Remarks to the Author):

This revised study examines perinatal exposure to maternal obesity and metabolic programming on offspring bronchial and pulmonary arterial smooth muscle thickening. The authors have provided significant new data that has strengthened the message and responded to many, but not all, of this reviewer's prior comments. Additional comments for consideration:

- 1. It would be helpful to present the parameters for pulmonary hypertension in a manner standard for the field, including right ventricular systolic pressure (not indexed for cardiac index) and the typical systemic blood pressure – or LV pressure as measured (if the aortic valve is normal), heart rate, cardiac output measures. In fact, these measurements are standard in the field and should be provided for all the mouse models since systemic hemodynamics can affect pulmonary hemodynamics. Furthermore, state-of-the-art methodology to assess bronchial and pulmonary vascular remodeling includes CT scans/reconstructions of the pulmonary vasculature to show distal pruning associated with pulmonary hypertension. While this may be beyond the scope of the current study, the same scan could also outline changes in the airways.**
- 2. There are several studies with mouse smooth muscle cell specific knockout models and in vitro studies using bronchial or pulmonary vascular smooth muscle cells. Since the smooth muscle knockout is global in the in vivo models and the disease model relies on the premise that the fat remote from the lungs has paracrine effects, you do need to show whether or not the same effects take place in the systemic vasculature and if there is vascular remodeling. Simply discussing kidney function is insufficient and doesn't address vascular remodeling. In addition, in vitro studies should be done with non-relevant cell types – non pulmonary vascular smooth muscle cells at the least – to demonstrate that the findings are either specific, or not specific, to smooth muscle cells in the pulmonary system.**
- 3. Please address the fact that the disease model is one of maternal obesity and early, but transient, obesity in mice yet the human GWAS studies are done in children age 10 with obesity. This doesn't address maternal obesity and in utero exposure. While the data presented are interesting, it could be considered a slightly different "model" and comparison.**

Reviewer #3 (Remarks to the Author):

The authors have adequately addressed all my concerns. I have no further comments.

Reviewer #4 (Remarks to the Author):

I'd like to thank the authors for having addressed some of my points. There are two points left to clarify:

- 1. "We are limited to using measures currently available in the UK Biobank since we did not have**

access to external summary statistics.”

I am not sure I'm following the authors remark: The application #13721, mentioned in the manuscript has access to all necessary data (it is an MR project with access to the full genetic data), so there is no limitation. Second, I do not understand why the authors have no access to external summary statistics, which ones? The ones are mentioned (by Neale lab) are all publicly available. Therefore, I do not see why the authors could not address my point to verify that sample overlap has not biased the results.

2. Further, the acknowledgements says “This research has been conducted using the UK Biobank Resource under Application Number 13721.” When looking up this approved project it showed that the PI is Themistocles Assimes, who has never been mentioned in the m/s and neither is an author of this paper. How is that possible?

We greatly appreciate the reviewers' comments. We revised the manuscript in accordance to the prior and recent comments. Please find our point-by-point reply to each comment below.

Reviewer #1 (Remarks to the Author):

[redacted]

Reviewer #2 (Remarks to the Author):

This revised study examines perinatal exposure to maternal obesity and metabolic programming on offspring bronchial and pulmonary arterial smooth muscle thickening. The authors have provided significant new data that has strengthened the message and responded to many, but not all, of this reviewer's prior comments. Additional comments for consideration:

1. It would be helpful to present the parameters for pulmonary hypertension in a manner standard for the field, including right ventricular systolic pressure (not indexed for cardiac index) and the typical systemic blood pressure – or LV pressure as measured (if the aortic valve is normal), heart rate, cardiac output measures. In fact, these measurements are standard in the field and should be provided for all the mouse models since systemic hemodynamics can affect pulmonary hemodynamics. Furthermore, state-of-the-art methodology to assess bronchial and pulmonary vascular remodeling includes CT scans/reconstructions of the pulmonary vasculature to show distal pruning associated with pulmonary hypertension. While this may be beyond the scope of the current study, the same scan could also outline changes in the airways.

We agree with the reviewer and included additional functional cardiac parameters. All measurements were performed blinded under a standardized isoflurane anesthesia with a constant flow of sustained 1-1.5% isoflurane in room air supplemented with 100 % O₂. Echocardiography was used to assess heart rate, cardiac output (CO), stroke volume (SV), right ventricular internal diameter (RVID), tricuspid annular plane systolic excursion (TAPSE), and left ventricular (LV) ejection fraction (EF). Invasive hemodynamic measurement was performed for right ventricular systolic pressure (RVSP) and left ventricular pressure (LVP). All parameters are shown in **Fig. 2G-O**. The low cardiac output and unaltered RVSP suggest an inability of the right ventricle to maintain the pressure. Furthermore, total pulmonary vascular resistance index (TPVRI) was significantly increased in offspring after perinatal obesity, reflecting high pulmonary resistance. These functional parameters together with vascular remodeling showing thickening of the smooth muscle cell layer indicate pulmonary hypertension after perinatal obesity. To exclude potential effects of perinatal HFD on the systemic hemodynamics we measured LVP by invasive catheterization and LV-EF by echocardiography and did not find any differences between offspring of high-fat diet (HFD)- and standard diet (SD)-fed dams, which again confirms primary effects on the pulmonary vasculature and subsequent influence on the right ventricle.

We absolutely concur that future studies should and will include more advanced imaging technologies, e.g. μ CT with reconstruction of the pulmonary vascular system, to investigate pulmonary vasculature and for example pruning.

2. There are several studies with mouse smooth muscle cell specific knockout models and in vitro studies using bronchial or pulmonary vascular smooth muscle cells. Since the smooth muscle knockout is global in the in vivo models and the disease model relies on the premise that the fat remote from the lungs has paracrine effects, you do need to show whether or not the same effects take place in the systemic vasculature and if there is vascular remodeling. Simply discussing kidney function is insufficient and doesn't address vascular remodeling. In addition, in vitro studies should be done with non-relevant cell types – non pulmonary vascular smooth muscle cells

at the least – to demonstrate that the findings are either specific, or not specific, to smooth muscle cells in the pulmonary system.

The reviewer points out a key aspect of obesity-related diseases. The underlying pathomechanisms are related to an inter-organ communication, including the impact of dysfunctional white adipose tissue along with subacute chronic inflammation as well as disturbed metabolism, e.g. hyperinsulinemia and hyperglycemia, on cell and organ function and ultimately on disease development.

Recent studies from our group showed that maternal HFD and perinatal obesity cause dysfunctional white adipose tissue (10), adversely effects kidney matrix and function (1), and induces hypothalamic inflammatory response in the offspring (11). In the current study, we show that perinatal obesity leads to bronchial and vascular remodeling through increased proliferation of SMCs. In order to test if this is a systemic effect we pursued two approaches:

First, we excised kidneys and liver from offspring of HFD- or SD-fed dams at postnatal day 21 (P21) and performed immunohistochemical staining for α SMA and von Willebrand factor (vWF), followed by assessment of the average vessel diameter per field of view along with the thickness of the SMC layer. The results show that perinatal obesity induces thickening of the SMC layer relative to vessel diameter in both kidney and liver ($p < 0.05$ and $p = 0.1$, respectively) (Suppl. Fig. 12A-D).

Second, we treated human aortic SMC (haSMC) with insulin or IL-6/soluble IL-6 receptor (IL-6+R) and found a significant activation of AKT and STAT3 signaling, respectively, along with increased proliferation. Moreover, we assessed pFoxO1 in haSMC after exposure to IL-6+R and demonstrate a nuclear-to-cytoplasmatic shift of pFoxO1 (Suppl. Fig. 12E-H). These findings are similar to our results in pulmonary SMC.

In summary, we provide initial *in vivo* and *in vitro* data that perinatal obesity has a systemic impact on vascular remodeling, possibly through an IL-6-mediated FoxO1 mechanism.

3. Please address the fact that the disease model is one of maternal obesity and early, but transient, obesity in mice yet the human GWAS studies are done in children age 10 with obesity. This doesn't address maternal obesity and in utero exposure. While the data presented are interesting, it could be considered a slightly different "model" and comparison.

We agree with the reviewer and discussed that the experimental murine model investigates perinatal (maternal and early-onset) obesity, whereas the children are characterized by their body weight at the age of 10 years and not by intrauterine or early postnatal exposure to obesity. This limitation has been included in the revised version of the manuscript (page 19; ll 507-510).

Reviewer #3 (Remarks to the Author):

The authors have adequately addressed all my concerns. I have no further comments.

We thank the reviewer.

Reviewer #4 (Remarks to the Author):

I'd like to thank the authors for having addressed some of my points. There are two points left to clarify:

1. "We are limited to using measures currently available in the UK Biobank since we did not have access to external summary statistics."

I am not sure I'm following the authors remark: The application #13721, mentioned in the manuscript has access to all necessary data (it is an MR project with access to the full genetic data), so there is no limitation. Second, I do not understand why the authors have no access to external summary statistics, which ones? The ones are mentioned (by Neale lab) are all publicly available. Therefore, I do not see why the authors could not address my point to verify that sample overlap has not biased the results.

We thank the reviewer for the comment, and we deeply agree about the biases that the sample overlapping may have produced in our MR results. When we previously talked about "external summary statistics", we referred to a GWAS summary statistics external to the UK Biobank and performed by a different consortium. The ideal design to perform this MR study is to use a GWAS summary statistic of childhood obesity not performed in the UK Biobank and the FEV1/FVC ratio performed in the UK Biobank as outcome and proxy for lung function. Unfortunately, we do not have access to a childhood obesity GWAS summary statistic external to the UK Biobank.

We followed the Reviewer's suggestion of randomly splitting the UK Biobank data into two groups and perform the MR analyses to avoid biases due to the winner's curse. But unfortunately, we did not replicate our results and we also detected a lack of statistical power. The reason why we did not replicate our results can be related to the lack of statistical power or, as the Reviewer suggested, the sample overlapping may have altered our results previously.

Consequently, we decided to remove our genetic and causal association analyses from the manuscript, and we focused our discussion on the observational analyses and on a previous publication performed by our group based on the causal association of birth weight and lung function.

Page 19 lines 504-506:

Our observational findings indicated a strong association of childhood obesity with reduced lung function. Consistent with our findings, previous Mendelian randomization analyses indicated a causal effect between birth weight and reduced lung function in adulthood (12).

2. Further, the acknowledgements says "This research has been conducted using the UK Biobank Resource under Application Number 13721." When looking up this approved project it showed that the PI is Themistocles Assimes, who has never been mentioned in the m/s and neither is an author of this paper. How is that possible?

We thank the reviewer for the comment. We completely agree with it. We used a wrong number by mistake. This research has been conducted using the UK Biobank Resource under Application Number

15860. The PI of this application is Dr. James Priest. He is a co-author and he contributed to the human analyses and to the drafting of the Manuscript.

We corrected the new version of the Manuscript accordingly at page 28 lines 738,739:

This research has been conducted using the UK Biobank Resource under Application Number 15860.

References:

1. P. Kasper *et al.*, Renal Metabolic Programming Is Linked to the Dynamic Regulation of a Leptin-Klf15 Axis and Akt/AMPKalpha Signaling in Male Offspring of Obese Dams. *Endocrinology* **158**, 3399-3415 (2017).
2. A. L. Ostermann *et al.*, Intestinal insulin/IGF1 signalling through FoxO1 regulates epithelial integrity and susceptibility to colon cancer. *Nat Metab* **1**, 371-389 (2019).
3. L. Sang, K. Kang, Y. Sun, Y. Li, B. Chang, FOXO4 ameliorates alcohol-induced chronic liver injury via inhibiting NF-kappaB and modulating gut microbiota in C57BL/6J mice. *Int Immunopharmacol* **96**, 107572 (2021).
4. W. Zhou *et al.*, FoxO4 inhibits NF-kappaB and protects mice against colonic injury and inflammation. *Gastroenterology* **137**, 1403-1414 (2009).
5. M. Zhu *et al.*, FoxO4 promotes early inflammatory response upon myocardial infarction via endothelial Arg1. *Circ Res* **117**, 967-977 (2015).
6. R. Savai *et al.*, Pro-proliferative and inflammatory signaling converge on FoxO1 transcription factor in pulmonary hypertension. *Nat Med* **20**, 1289-1300 (2014).
7. H. M. Al-Tamari *et al.*, FoxO3 an important player in fibrogenesis and therapeutic target for idiopathic pulmonary fibrosis. *EMBO Mol Med* **10**, 276-293 (2018).
8. P. Rao *et al.*, Promotion of beta-catenin/Foxo1 signaling ameliorates renal interstitial fibrosis. *Lab Invest* **99**, 1689-1701 (2019).
9. X. Pan, Y. Zhang, H. G. Kim, S. Liangpunsakul, X. C. Dong, FOXO transcription factors protect against the diet-induced fatty liver disease. *Sci Rep* **7**, 44597 (2017).
10. T. Litzenburger *et al.*, Maternal high-fat diet induces long-term obesity with sex-dependent metabolic programming of adipocyte differentiation, hypertrophy and dysfunction in the offspring. *Clin Sci (Lond)*, (2020).
11. E. Rother *et al.*, Hypothalamic JNK1 and IKKbeta activation and impaired early postnatal glucose metabolism after maternal perinatal high-fat feeding. *Endocrinology* **153**, 770-781 (2012).
12. C. Kuiper-Makris *et al.*, Mendelian randomization and experimental IUGR reveal the adverse effect of low birth weight on lung structure and function. *Sci Rep* **10**, 22395 (2020).

REVIEWER COMMENTS

Reviewer #1 (Remarks to the Author):

[redacted]

Reviewer #2 (Remarks to the Author):

The authors have been very responsive to prior comments and the newly added data are of interest. Two remaining comments (first wasn't addressed on this revision):

1. Please add the systemic blood pressure for the animal models. While the LVSP approximates the systemic blood pressure, the diastolic blood pressure can't be ascertained. Since diastolic hypertension is associated with obesity it would be a good data point. If not available, list as a limitation.
2. Fig Supple 12 - please identify which stain is vWF and which is aSMA. It's not provided in the figure or the legend.

Reviewer #4 (Remarks to the Author):

The authors have addressed my criticism by removing the whole MR analysis. It is a pity that split-sample results were contradicting the full sample ones.

We greatly appreciate the reviewers' comments. We revised the manuscript in accordance to the prior and recent comments. Please find our point-by-point reply to each comment below.

Reviewer #1 (Remarks to the Author):

[redacted]

Reviewer #2 (Remarks to the Author):

1. Please add the systemic blood pressure for the animal models. While the LVSP approximates the systemic blood pressure, the diastolic blood pressure can't be ascertained. Since diastolic hypertension is associated with obesity it would be a good data point. If not available, list as a limitation.

We appreciate the comment. We have included the systolic blood pressure (SBP) and the diastolic blood pressure (DBP) in Figure 2.

2. Fig Supple 12 - please identify which stain is vWF and which is aSMA. It's not provided in the figure or the legend.

We revised supplementary figure 12.

References:

1. R. Savai *et al.*, Pro-proliferative and inflammatory signaling converge on FoxO1 transcription factor in pulmonary hypertension. *Nat Med* **20**, 1289-1300 (2014).
2. A. L. Ostermann *et al.*, Intestinal insulin/IGF1 signalling through FoxO1 regulates epithelial integrity and susceptibility to colon cancer. *Nat Metab* **1**, 371-389 (2019).